# Members of the ELMOD protein family specify formation of distinct aperture domains on the *Arabidopsis* pollen surface

**Yuan Zhou, Prativa Amom[†], Sarah H Reeder, Byung Ha Lee[‡], Adam Helton[§], Anna A Dobritsa\***

Department of Molecular Genetics and Center for Applied Plant Sciences, Ohio State University, Columbus, United States

**\*For correspondence:**
dobritsa.1@osu.edu

**Present address:** [†]Cincinnati Children's Hospital Medical Center, University of Cincinnati, Cincinnati, United States; [‡]Macrogen, Inc, Seoul, South Korea; [§]PPD Laboratories, Middleton, United States

**Competing interest:** The authors declare that no competing interests exist.

**Abstract** Pollen apertures, the characteristic gaps in pollen wall exine, have emerged as a model for studying the formation of distinct plasma membrane domains. In each species, aperture number, position, and morphology are typically fixed; across species they vary widely. During pollen development, certain plasma membrane domains attract specific proteins and lipids and become protected from exine deposition, developing into apertures. However, how these aperture domains are selected is unknown. Here, we demonstrate that patterns of aperture domains in *Arabidopsis* are controlled by the members of the ancient ELMOD protein family, which, although important in animals, has not been studied in plants. We show that two members of this family, MACARON (MCR) and ELMOD_A, act upstream of the previously discovered aperture proteins and that their expression levels influence the number of aperture domains that form on the surface of developing pollen grains. We also show that a third ELMOD family member, ELMOD_E, can interfere with MCR and ELMOD_A activities, changing aperture morphology and producing new aperture patterns. Our findings reveal key players controlling early steps in aperture domain formation, identify residues important for their function, and open new avenues for investigating how diversity of aperture patterns in nature is achieved.

## Introduction

As part of cell morphogenesis, cells often form distinct plasma membrane domains that acquire specific combinations of proteins, lipids, and extracellular materials. Yet how these domains are selected and specified is often unclear. Pollen apertures offer a powerful model for studying this process. Apertures are the characteristic gaps on the pollen surface that receive little to no deposition of the pollen wall exine; during their formation, certain regions of the plasma membrane are selected and specified as aperture domains (*Zhou and Dobritsa, 2019*). Pollen apertures create some of the most recognizable patterns on the pollen surface, usually conserved within a species but highly variable across species (*Furness and Rudall, 2004*). For instance, in wild-type *Arabidopsis* pollen, apertures are represented by three long and narrow furrows, equally spaced on the pollen surface and oriented longitudinally (*Figure 1A and A'*). In other species, aperture positions, number, and morphologies can be different, suggesting that the mechanisms guiding aperture formation are diverse. While the diversity of aperture patterns has captivated scientists for decades (*Furness and Rudall, 2004*; *Matamoro-Vidal et al., 2016*; *Walker, 1974*; *Wodehouse, 1935*), studies of the associated molecular mechanisms have only recently begun (*Dobritsa and Coerper, 2012*; *Dobritsa et al., 2018*; *Lee et al., 2018*; *Reeder et al., 2016*; *Zhang et al., 2020*).

**eLife digest** Zooming in on cells reveals patterns on their outer surfaces. These patterns are actually a collection of distinct areas of the cell surface, each containing specific combinations of molecules. The outer layers of pollen grains consist of a cell wall, and a softer cell membrane that sits underneath. As a pollen grain develops, it recruits certain fats and proteins to specific areas of the cell membrane, known as 'aperture domains'. The composition of these domains blocks the cell wall from forming over them, leading to gaps in the wall called 'pollen apertures'. Pollen apertures can open and close, aiding reproduction and protecting pollen grains from dehydration.

The number, location, and shape of pollen apertures vary between different plant species, but are consistent within the same species. In the plant species *Arabidopsis thaliana*, pollen normally develops three long and narrow, equally spaced apertures, but it remains unclear how pollen grains control the number and location of aperture domains.

Zhou et al. found that mutations in two closely related *A. thaliana* proteins – ELMOD_A and MCR – alter the number and positions of pollen apertures. When *A. thaliana* plants were genetically modified so that they would produce different levels of ELMOD_A and MCR, Zhou et al. observed that when more of these proteins were present in a pollen grain, more apertures were generated on the pollen surface. This finding suggests that the levels of these proteins must be tightly regulated to control pollen aperture numbers. Further tests revealed that another related protein, called ELMOD_E, also has a role in domain formation. When artificially produced in developing pollen grains, it interfered with the activity of ELMOD_A and MCR, changing pollen aperture shape, number, and location.

Zhou et al. identified a group of proteins that help control the formation of domains in the cell membranes of *A. thaliana* pollen grains. Further research will be required to determine what exactly these proteins do to promote formation of aperture domains and whether similar proteins control domain development in other organisms.

Aperture domains first become visible at the tetrad stage of pollen development, when four sister microspores, the products of meiosis, are held together under the common callose wall and aperture factors, such as INAPERTURATE POLLEN1 (INP1) and D6 PROTEIN KINASE-LIKE3 (D6PKL3) in *Arabidopsis* and OsINP1 and DEFECTIVE IN APERTURE FORMATION1 (OsDAF1) in rice, accumulate at distinct domains of the microspore plasma membranes (*Dobritsa and Coerper, 2012*; *Dobritsa et al., 2018*; *Lee et al., 2018*; *Zhang et al., 2020*). These domains become protected from exine deposition and develop into apertures (*Dobritsa et al., 2018*; *Zhang et al., 2020*). Yet how aperture domains are selected and what mechanism guides their patterning remain completely unknown.

Recently, we isolated a new *Arabidopsis* mutant, *macaron* (*mcr*), in which pollen, instead of forming three apertures, develops a single ring-shaped aperture, suggesting that the affected gene is involved in specifying positions and number of aperture domains (*Plourde et al., 2019*). Here, we perform a detailed analysis of this mutant and identify the *MCR* gene. We demonstrate that it belongs to the ancient family of ELMOD proteins, and that together with another member of this protein family in *Arabidopsis*, ELMOD_A, MCR acts at the beginning of the aperture formation pathway as a positive regulator of aperture domain specification. We provide evidence that aperture domains are highly sensitive to the levels of MCR and ELMOD_A, which can positively or negatively affect their number. We further demonstrate that a third member of this family, ELMOD_E, has an ability to influence the number, positions, and morphology of aperture domains, and we identify specific protein residues critical for this ability. Our study elucidates key molecular factors controlling aperture patterning and functionally characterizes members of the widespread, yet thus far uncharacterized family of the plant ELMOD proteins.

## Results

### *mcr* mutants develop a single ring-shaped pollen aperture composed of two equidistantly placed longitudinal apertures

In a screen of EMS-mutagenized *Arabidopsis* plants, we discovered four non-complementing mutants, which, instead of three equidistant pollen apertures, produced a single ring-shaped aperture dividing

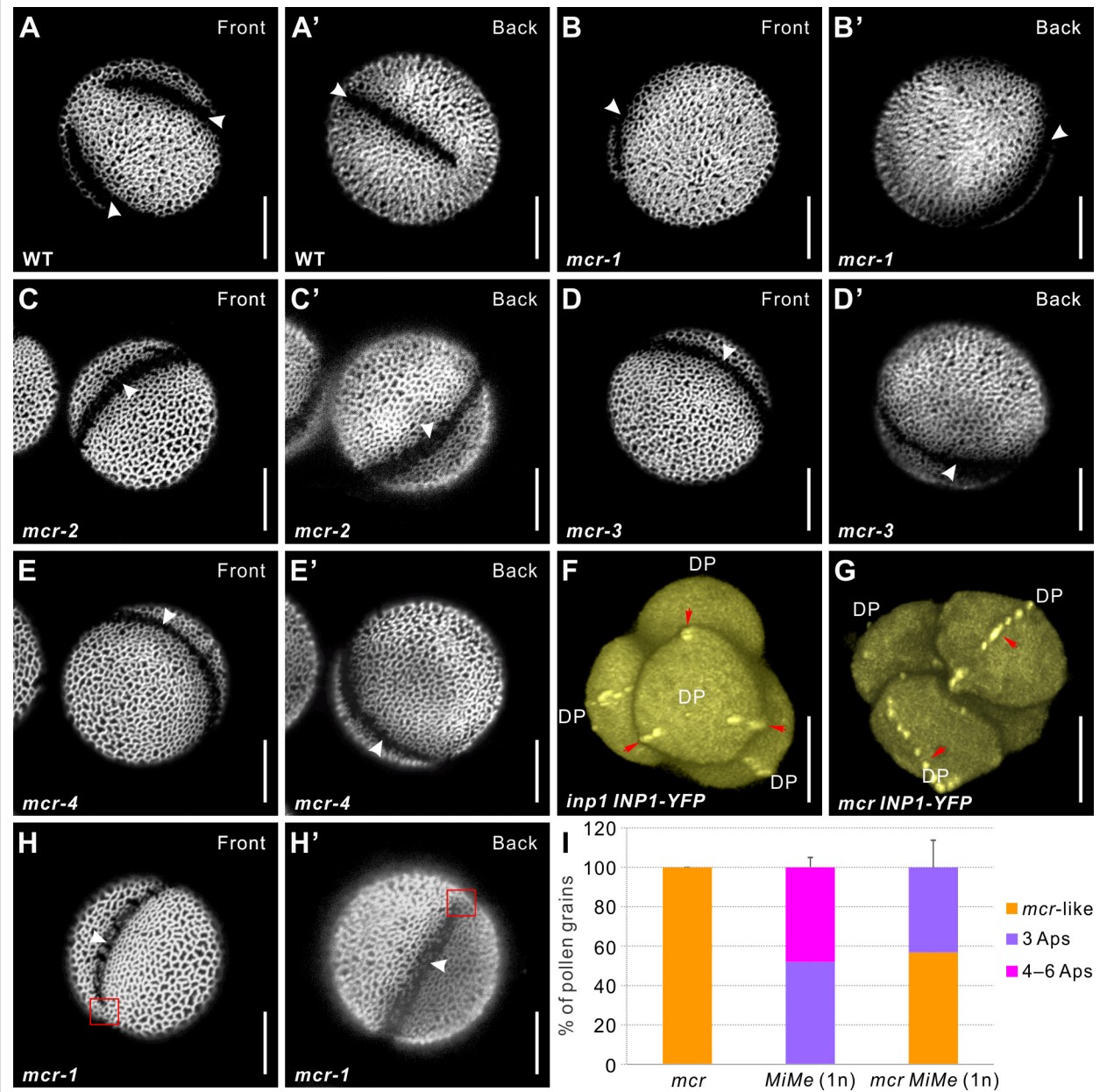

**Figure 1.** Mutations in *MCR* reduce aperture number. (**A–E′**) Confocal images of auramine O-stained pollen grains from wild-type (L*er*) and four *mcr* EMS mutants. Front (α) and back (α′) show the opposite views of the same pollen grain here and in other figures as indicated. (**F, G**) 3D reconstructions of tetrad-stage microspores showing lines of INP1-YFP (red arrows) in *inp1* and *mcr* mutants. DP: distal pole. (**H, H′**) *mcr* pollen with two apertures. Red boxes mark the regions where apertures are not fused. (**I**) Percentage of pollen grains with indicated number of apertures in pollen populations from *mcr*, 1n *MiMe*, and 1n *mcr MiMe* plants (n = 75–500). Error bars represent SD, calculated from 4 to 6 independent biological replicates. Apertures are indicated with arrowheads in (**A–E′**) and (**H, H′**). Scale bars, 10 μm.

The online version of this article includes the following figure supplement(s) for figure 1:

**Figure supplement 1.** Diagrams summarizing the INP1-YFP localization in *inp1* and *mcr* tetrads, based on confocal imaging and 3D reconstruction of *DMC1pr:INP1-YFP*-expressing tetrads.

**Figure supplement 2.** The reducing effect of *mcr* mutations on aperture number is manifested across different ploidy levels and arrangements of microspores.

each pollen grain into two equal parts (*Figure 1B-E'*). As the mutant phenotype resembled the French meringue dessert, we named these mutations *macaron* (alleles *mcr-1* through *mcr-4*).

Imaging of *mcr* microspore tetrads demonstrated that they develop normally and achieve a regular tetrahedral conformation. The ring-shaped aperture domains in *mcr* microspores, visualized with the help of the reporter INP1-YFP, are positioned so that they pass through the proximal and distal poles of each microspore (*Figure 1G*; compare with the INP1-YFP localization in the absence of *mcr* mutation in *Figure 1F*). Thus, like in wild-type pollen, apertures in *mcr* are placed longitudinally. However, while aperture positions in each wild-type microspore are coordinated with aperture positions in its three sisters (*Dobritsa et al., 2018*; *Reeder et al., 2016*), in *mcr*, the ring-shaped apertures appear to be placed independently in sister microspores (*Figure 1—figure supplement 1*). Occasionally, instead of ring-shaped apertures, *mcr* pollen displays two unconnected apertures (*Figure 1H and H'*), suggesting that the ring-shaped aperture is a product of a two-aperture fusion. Thus, *mcr* mutations reduce the number of apertures, but do not affect their furrow morphology, longitudinal orientation, and equidistant placement. Like all the other previously characterized pollen aperture mutants in *Arabidopsis*, including *inp1* and *inp2*, which completely lack apertures (*Dobritsa and Coerper, 2012*; *Dobritsa et al., 2011*; *Lee et al., 2021*), *mcr* mutants exhibited no obvious fertility defects.

## The *mcr* mutation reduces aperture number across different levels of ploidy and arrangements of microspores

We previously showed that aperture number strongly depends on microspore ploidy and is sensitive to cytokinetic defects that disrupt formation of normal tetrahedral tetrads, creating other arrangements of post-meiotic microspores (*Reeder et al., 2016*). While normal haploid (1n) pollen develops three apertures, diploid (2n) pollen produces either four or a mixture of four and six apertures, depending on whether it was generated through tetrads or dyads. In contrast, 2n *mcr* pollen, produced through either tetrads or dyads, has three equidistant apertures (*Plourde et al., 2019*), suggesting that the increasing effect of higher ploidy on aperture number is counterbalanced by the defect in the *MCR* function.

We have now extended this analysis by assessing the effects of the *mcr* mutation on aperture formation under additional perturbations of ploidy or post-meiotic microspore arrangement. By creating 1n *Mitosis instead of Meiosis* (*MiMe*) plants (*d'Erfurth et al., 2009*) with the *mcr* mutation, we generated *mcr* pollen with normal ploidy (1n) via dyads, and not tetrads. As shown previously (*Reeder et al., 2016*), a majority of the 1n *MiMe* pollen grains (~60%) develop three normal apertures, with the rest forming mostly six apertures (*Figure 1I*, *Figure 1—figure supplement 2A–C'*). Yet, in the pollen of the 1n *mcr MiMe* plants, the number of apertures was reduced, with ~50–70% of pollen developing the *mcr* phenotype (either ring-shaped or two apertures) and the rest forming three apertures (*Figure 1I*, *Figure 1—figure supplement 2D–E'*).

We further perturbed microspore formation and ploidy by crossing *mcr-1* with a mutant defective in the *TETRASPORE (TES)* gene. In *tes* mutants, microspore mother cells (MMCs) go through meiosis but fail to undergo cytokinesis, producing large pollen grains with four haploid nuclei and a high number (~10 or more) of irregularly placed and fused apertures (*Reeder et al., 2016*; *Spielman et al., 1997*). Although in the double *mcr tes* mutant apertures are often positioned irregularly and fused together, their number was usually lower (~4–6) than in the single *tes* mutant (*Figure 1—figure supplement 2F–G'*). Altogether, these results indicate that *mcr* mutations have an overall reducing effect on aperture number, manifested across different levels of pollen ploidy and post-meiotic microspore arrangements.

## *mcr* acts genetically upstream of the aperture factors *INP1* and *D6PKL3*

In wild-type tetrad-stage microspores, aperture factors INP1 and D6PKL3 localize to the three longitudinal aperture domains of the plasma membrane (*Dobritsa and Coerper, 2012*; *Dobritsa et al., 2018*; *Lee et al., 2018*). Since *mcr* mutation affects INP1-YFP localization, causing it to migrate to a ring-shaped membrane domain (*Figure 1G*), we tested whether *mcr* also affects the localization of D6PKL3, which likely acts upstream of INP1. We introgressed the previously characterized transgenic reporter *D6PKL3pr:D6PKL3-YFP* (*Lee et al., 2018*) into the *mcr-1* background. In *mcr* microspores,

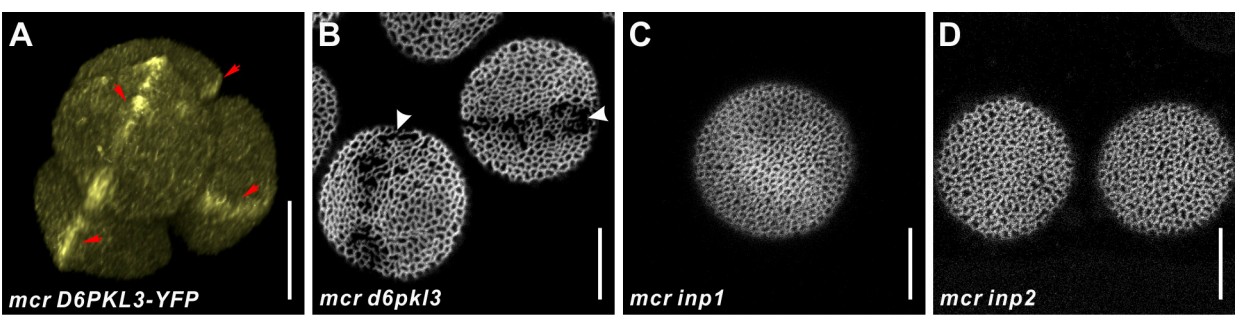

**Figure 2.** *MCR* acts genetically upstream of the three known aperture factors, *D6PKL3*, *INP1*, and *INP2*. (**A**) 3D reconstruction of tetrad-stage microspores showing lines of D6PKL3-YFP in *mcr* tetrads. (**B–D**) Pollen grains of *mcr d6pkl3*, *mcr inp1*, and *mcr inp2* double mutants. Apertures are indicated with arrowheads, and D6PKL3-YFP lines are indicated with red arrows. Scale bars, 10 μm.

D6PKL3-YFP re-localized to a single ring-shaped domain (*Figure 2A*), indicating that MCR acts upstream of both INP1 and D6PKL3.

We also examined the genetic interactions between *MCR* and other aperture factors, including the recently discovered *INP2* (*Lee et al., 2021*), by combining their mutations. *d6pkl3* single mutants develop three apertures partially covered by exine (*Lee et al., 2018*). Pollen of the *mcr d6pkl3* double mutants developed single ring-shaped apertures that were partially covered by exine, indicating that the two genes have an additive effect on aperture phenotype (*Figure 2B*). In contrast, pollen grains of *mcr inp1* and *mcr inp2* completely lacked apertures, phenocopying single *inp1* and *inp2* mutants (*Dobritsa and Coerper, 2012*; *Lee et al., 2021*; *Figure 2C and D*). This indicates that *INP1* and *INP2* are epistatic to *MCR*, consistent with their roles as factors absolutely essential for aperture formation.

## MCR is a member of the ancient ELMOD protein family

We mapped the *mcr-1* defect to a 77 kb interval on the second chromosome. One of the 25 genes in this interval, *At2g44770*, had a C-to-T mutation converting a highly conserved Pro165 (see below) into a Ser (*Figure 3A*, *Figure 3—figure supplement 1*). Sequencing of *At2g44770* from the other three *mcr* alleles also revealed mutations (*Figure 3A*, *Figure 3—figure supplement 1*). *mcr-2* had a G-to-A mutation converting Gly129 into an Asp. *mcr-3* had a G-to-A mutation affecting the last nucleotide of the fifth intron, disrupting the splice acceptor site and causing a frame shift in the middle of the critical catalytic region (see below). In *mcr-4*, no mutations in the coding sequence (CDS) of *At2g44770* were found; however, there was a G-to-A mutation 310 nt downstream of the stop codon in its 3′ untranslated region (3′ UTR), suggesting that the 3′ UTR is important for regulation of this gene (*Figure 3A*). In addition, plants with T-DNA insertions in this gene (*mcr-5*, *mcr-6*, and *mcr-7*) all produced pollen with the *mcr* phenotype (*Figure 3A*, *Figure 3—figure supplement 2*). Yet the T-DNA mutations, likely due to their residence in introns, were hypomorphic, as some pollen with three normal apertures was found in their populations (9% in *mcr-5* [n = 179], 13% in *mcr-6* [n = 216], and 22% in *mcr-7* [n = 78]). This was in contrast to the *mcr-1* through *mcr-4* mutations, in which the *mcr* aperture phenotype was fully penetrant.

We further verified the identity of *MCR* as *At2g44770* by creating complementation constructs and expressing them in the *mcr-1* mutant. The genomic construct *MCRpr:gMCR* (driven by the 3 kb DNA fragment upstream of the start codon [referred to as the *MCR* promoter] and containing introns and the 0.8 kb region downstream of the stop codon) restored three normal apertures in 10/10 $T_1$ transgenic plants (*Figure 3B and B′*). A similar genomic construct expressing protein fused at the C-terminus with yellow fluorescent protein (YFP) also successfully restored apertures (*Figure 3C and C′*). In contrast, the *MCRpr:MCR CDS* construct, which contained only the CDS driven by the *MCR* promoter, did not rescue the *mcr* phenotype (0/6 $T_1$ plants had three apertures restored; *Figure 3D and D′*), indicating that additional regulatory regions are required for expression of this gene, consistent with the notion of the 3′ UTR importance. The *MCR* promoter and 3′ UTR were then included in all constructs for which we sought *MCR*-like expression and are herein referred to as the *MCR* regulatory regions.

The protein encoded by *At2g44770* contains the Engulfment and Cell Motility (ELMO) domain (InterPro006816) (*Figure 3A*, *Figure 3—figure supplement 1*). In animals, proteins with this domain

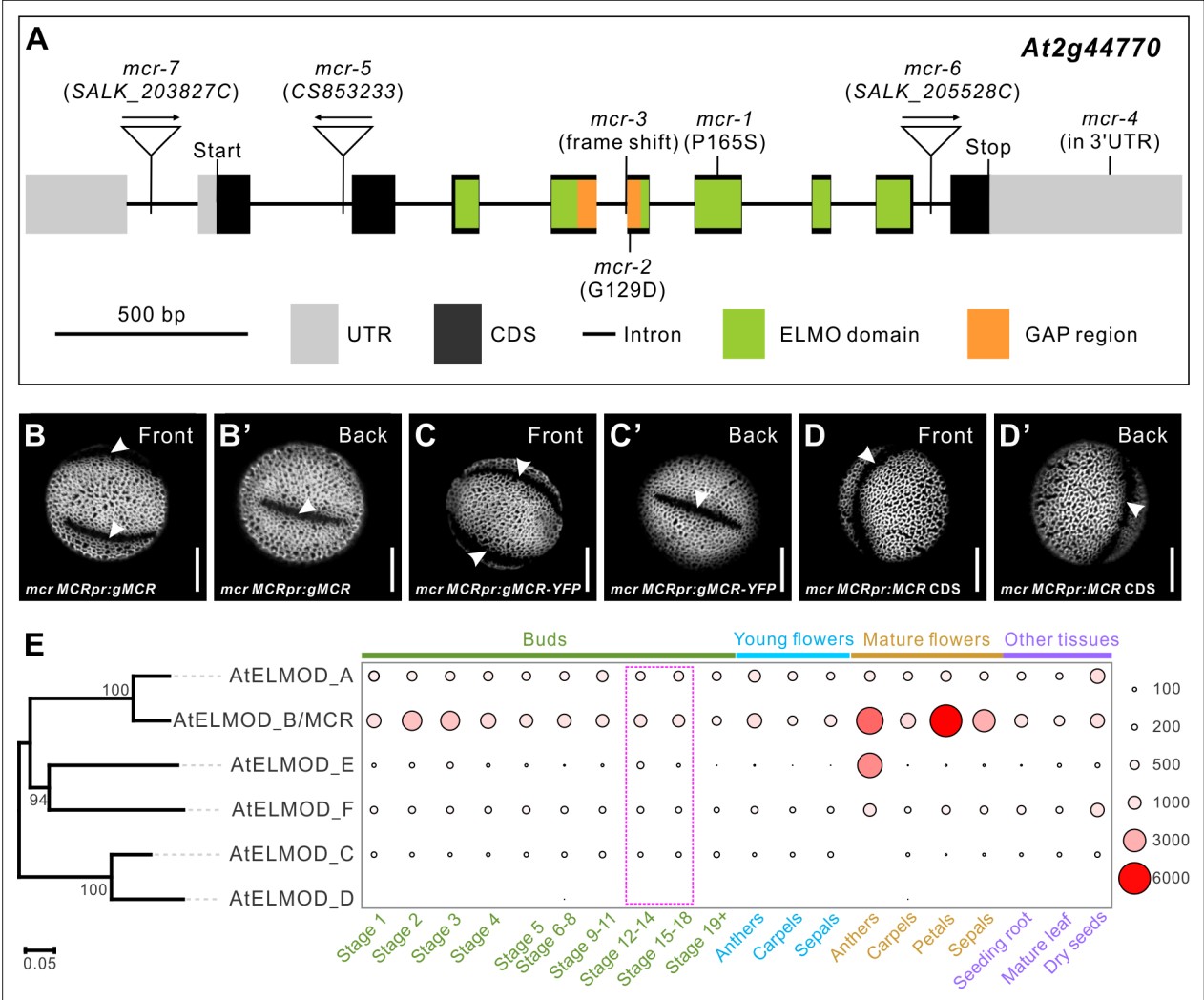

**Figure 3.** MCR, a member of the ELMOD protein family, is encoded by *At2g44770*. (**A**) Diagram of the *MCR* gene (*At2g44770*). Positions of seven mutations and several gene and protein regions are indicated. (**B–D'**) Pollen grains from *mcr* plants expressing *MCRpr:gMCR*, *MCRpr:gMCR-YFP*, and *MCRpr:MCR CDS* constructs. Apertures are indicated with arrowheads. Scale bars, 10 μm. (**E**) Phylogenetic tree of the *Arabidopsis* ELMOD proteins and expression patterns of the corresponding genes. Bootstrap values (%) for 1000 replicates are shown at tree nodes. RNA-seq data obtained from the TRAVA database are presented as a bubble heatmap (values indicate normalized read counts). Magenta box marks the bud stages associated with pollen aperture formation (stages follow the TRAVA nomenclature).

The online version of this article includes the following figure supplement(s) for figure 3:

**Figure supplement 1.** Protein sequence alignment of *Arabidopsis* ELMOD proteins.

**Figure supplement 2.** T-DNA insertion mutants of *MCR* produce pollen with a single ring-shaped aperture.

belong to two families: (1) smaller ELMOD proteins, containing only the ELMO domain, and (2) larger ELMO proteins, containing, besides the ELMO domain, several other protein domains (*East et al., 2012*). The ELMOD family is believed to be the more ancient of the two, with ELMOD proteins already present in the last common ancestor of all eukaryotes and ELMO proteins appearing later in evolution in the opisthokont clade (*East et al., 2012*). In mammals, ELMOD proteins act as non-canonical GTPase activating proteins (GAPs) for regulatory GTPases of the ADP-ribosylation factor (Arf) family, a subgroup within the Ras superfamily that includes Arf and Arf-like (Arl) proteins (*Bowzard et al., 2007*; *Ivanova et al., 2014*; *Turn et al., 2020*). Unlike animals, plants only have members of the ELMOD family, and their roles remain essentially uncharacterized.

## Another member of the *Arabidopsis* ELMOD family, ELMOD_A, is also involved in aperture formation

In *Arabidopsis*, the ELMOD family consists of six members, ELMOD_A through ELMOD_F (*Figure 3E*, *Figure 3—figure supplement 1*), in the nomenclature of *East et al., 2012*. MCR is ELMOD_B. One of the other five proteins, ELMOD_A, shares 86% sequence identity with MCR, and the rest have ~50–55% sequence identity with both MCR and ELMOD_A. Although the ELMOD proteins are broadly expressed in *Arabidopsis*, young buds at or near the stages when apertures develop express mostly MCR and ELMOD_A (*Figure 3E*).

Given the high similarity between MCR and ELMOD_A, we wondered if ELMOD_A also aids in aperture formation. We disrupted ELMOD_A with CRISPR/Cas9 by introducing a single-nucleotide insertion (A) after the codon 64 (*Figure 4A*). Although this created a shift in the open reading frame and an early stop codon, it did not affect aperture formation (*Figure 4B and B'*). (Likewise, a second *elmod_a* CRISPR allele, in which deletion of the last nucleotide of the codon 64 resulted in a different frame shift and creation of another early stop codon, did not affect the formation of apertures.) We hypothesized that the lack of phenotype could be due to the ELMOD_A redundancy with MCR. To test this, we crossed the *elmod_a* mutant (carrying the CRISPR/Cas9 transgene) with the *mcr-1* mutant. Already in the $F_1$ generation, when all plants were expected to be double heterozygotes, we found several plants producing pollen with the *mcr*-like aperture phenotype (*Figure 4C and C'*). Sequencing of the *MCR* and *ELMOD_A* genes from these plants showed that, as expected, they were heterozygous for *MCR*; however, they had homozygous or biallelic mutations in *ELMOD_A*, indicating that the CRISPR/Cas9 transgene continued targeting the wild-type copy of *ELMOD_A* in the $F_1$ progeny of the cross.

The phenotype of these *mcr/+ elmod_a* mutants revealed that in the absence of *ELMOD_A*, *MCR* displays haploinsufficiency. Notably, when at least one wild-type copy of *ELMOD_A* is present, *MCR* is haplosufficient (*Figure 4D and D'*). Therefore, these paralogs play redundant roles in the formation of aperture domains. Yet, since MCR can specify three normal apertures in the absence of ELMOD_A but not vice versa, its role appears to be more prominent compared to that of ELMOD_A.

We also tested how the lack of one wild-type copy of *ELMOD_A* and both wild-type copies of *MCR*, as well as the lack of wild-type copies of both genes, would affect aperture formation. In the *mcr elmod_a/+* plants, pollen had the *mcr* phenotype (*Figure 4E and E'*). However, when the function of both genes was completely disrupted, the resulting pollen produced either one greatly disrupted aperture with an abnormal, circular morphology and partially covered with exine, or formed no apertures (*Figure 4F-G'*). Thus, the simultaneous loss of function of the two *ELMOD* family genes has a synergistic effect on aperture formation.

To confirm that these defects were caused by mutations in *ELMOD_A* and not off-site CRISPR targeting events, as well as to identify the *ELMOD_A* regulatory regions, we created two *ELMOD_A* genomic constructs driven by the 2 kb region upstream of its start codon – *EApr:gELMOD_A* (which also included a 0.3 kb *ELMOD_A* 3′ UTR) and *EApr:gELMOD_A-YFP* (tagged with YFP and lacking the *ELMOD_A* 3′ UTR) – and transformed them into the *mcr elmod_a* double mutant, which no longer carried the CRISPR/Cas9 transgene. Both constructs successfully rescued formation of apertures (5/5 and 31/33 $T_1$ plants, respectively, *Figure 4H–I'*), indicating that the selected promoter region is sufficient for *ELMOD_A* functional expression. In addition, when *ELMOD_A* was expressed in the *mcr* single mutant from either its own promoter or from the *MCR* regulatory regions (*MCRpr:gELMOD_A--YFP-MCR3'UTR*), it also complemented the loss of *MCR* (12/12 and 14/14 $T_1$ plants; *Figure 4J–K'*).

Thus, both ELMOD_A and MCR participate in aperture domain specification. Formation of three apertures in *Arabidopsis* pollen requires either two intact copies of *MCR* or at least one copy of each of these two ELMOD family members.

## MCR and ELMOD_A are expressed in the developing pollen lineage but, unlike other aperture factors, do not accumulate at the aperture membrane domains

According to the publicly available RNA-seq data (*Klepikova et al., 2016*), *MCR* and *ELMOD_A* are both expressed in young buds with pollen at or near the tetrad stage of development (*Figure 3E*). To confirm that in these buds *MCR* and *ELMOD_A* are expressed in the developing pollen lineage, we created transcriptional reporter constructs *MCRpr:H2B-RFP* and *EApr:H2B-RFP*, expressing the

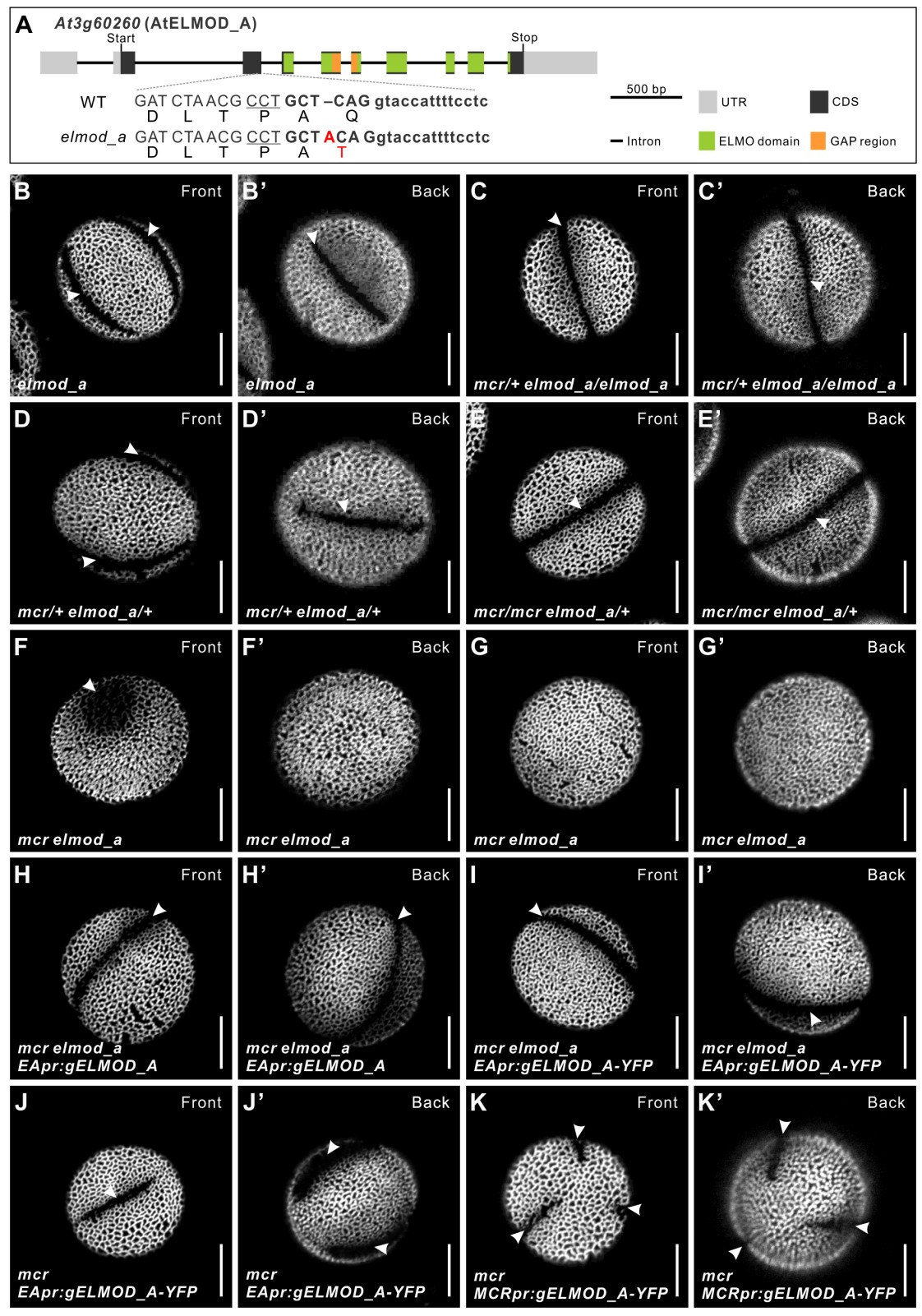

**Figure 4.** ELMOD_A is involved in aperture formation. (**A**) Diagram of the *ELMOD_A* gene (*At3g60260*) and the CRISPR/Cas9-induced *elmod_a* mutation. Nucleotide and amino acid changes are indicated with red capital letters. 20 bp target sequence next to the underlined protospacer adjacent motif is shown in bold. Lowercase letters represent sequence of an intron. (**B–G′**) Pollen grains from *elmod_a* mutant and from the indicated homo- and heterozygous combinations of *elmod_a* and *mcr* mutations. (**H–I′**) Pollen grains from *mcr elmod_a* plants expressing *EApr:gELMOD_A*

*Figure 4 continued on next page*

*Figure 4 continued*

and *EApr:gELMOD_A-YFP* constructs. (**J–K'**) Pollen grains from *mcr* plants expressing *EApr:gELMOD_A-YFP* and *MCRpr:gELMOD_A-YFP* constructs. Apertures are indicated with arrowheads. Scale bars, 10 μm.

nuclear marker H2B tagged with red fluorescent protein and transformed them into wild-type *Arabidopsis*. In the resulting transgenic lines, *MCR* and *ELMOD_A* promoters were active in the developing pollen lineage (MMCs, tetrads, and young free microspores) as well as in somatic anther layers (*Figure 5A*).

To find out if, like the previously discovered aperture factors INP1 and D6PKL3, MCR and ELMOD_A accumulate at the aperture domains of tetrad-stage microspores, we determined the subcellular localization of the YFP-tagged proteins expressed from the translational reporters *MCRpr:gMCR-YFP* and *EApr:gELMOD_A-YFP*, which rescued mutant phenotypes. Consistent with the results from the transcriptional reporters, the YFP signal was present in MMCs, tetrads, and young microspores (*Figure 5B and C*). This signal was diffusely localized in the cytoplasm and prominently enriched in the nucleoplasm. No specific enrichment near the plasma membrane was observed. Therefore, MCR and ELMOD_A specify positions and number of aperture domains without visibly congregating there.

## Invariant arginine in the putative GAP region is essential for MCR and ELMOD_A functions

Although ELMOD proteins do not have the typical GAP domain associated with the canonical Arf GAP proteins, they contain a conserved stretch of 26 amino acids, with 13 residues exhibiting a particularly high degree of conservation and forming the consensus sequence $WX_3G(F/W)QX_3PXTD(F/L)$**R**$GX$-$GX_3LX_2L$. In mammalian ELMODs, this region is proposed to mediate their Arf/Arl GAP activity (*East et al., 2012*). The presence of the invariant Arg in this region is of particular importance since the activity of many GAP proteins of the Ras GTPase superfamily, including canonical Arf GAPs, relies on a catalytic Arg (*Scheffzek et al., 1998*). Indeed, in mammalian ELMODs, the Arg in this putative GAP region was shown to be essential for their GAP activity, consistent with its role as the catalytic residue (*East et al., 2012*). Even relatively small changes at this position, such as conversion to Lys, resulted in the complete loss of GAP activity.

Although plant ELMODs have only limited similarity to mammalian proteins (e.g., the *Arabidopsis* and human ELMODs have ~20% sequence identity), they contain the same conserved region and invariant Arg residue (*Figure 6A*). To test if this region is essential for function in MCR and ELMOD_A, we created constructs in which the invariant Arg (R127) was substituted with Lys (*MCRpr:gMCR$^{R127K}$-YFP* and *EApr:gELMOD_A$^{R127K}$-YFP*). These constructs were then expressed, respectively, in the *mcr* and *mcr elmod_a* mutants. Unlike the constructs with the wild-type MCR and ELMOD_A, the R127K constructs, although expressed normally, completely failed to restore the expected aperture patterns (0/8 $T_1$ plants for MCR$^{R127K}$; 0/12 T1 plants for ELMOD_A$^{R127K}$), indicating that, like in mammalian ELMODs, the Arg in the putative GAP region is critical for the activity of MCR and ELMOD_A (*Figure 6B and C*).

## The number of developing aperture domains is highly sensitive to the levels of MCR and ELMOD_A

While working with *MCR-YFP* and *ELMOD_A-YFP* transgenic lines, we made a surprising discovery. We noticed that while most of these lines had apertures restored to the expected number (i.e., three apertures for *mcr MCRpr:gMCR-YFP* and a ring-shaped aperture/two apertures for *mcr elmod_a EApr:ELMOD_A-YFP*), in some transgenic $T_1$ lines the number of apertures exceeded the expectations: with up to six apertures forming in *mcr MCRpr:gMCR-YFP* and up to four apertures in *mcr elmod_a EApr:gELMOD_A-YFP* (*Figure 7A and B'*).

To test if different aperture numbers could be due to different levels of transgene expression, we examined YFP fluorescence in homozygous lines producing different aperture numbers. For both *MCR* and *ELMOD_A* transgenes, the number of apertures positively correlated with the level of YFP signal in the microspore cytoplasm and nucleoplasm (*Figure 7C–E*, *Figure 7—figure supplement 1A*). In addition, in some lines, the number of apertures further increased in $T_2$ or $T_3$ generations

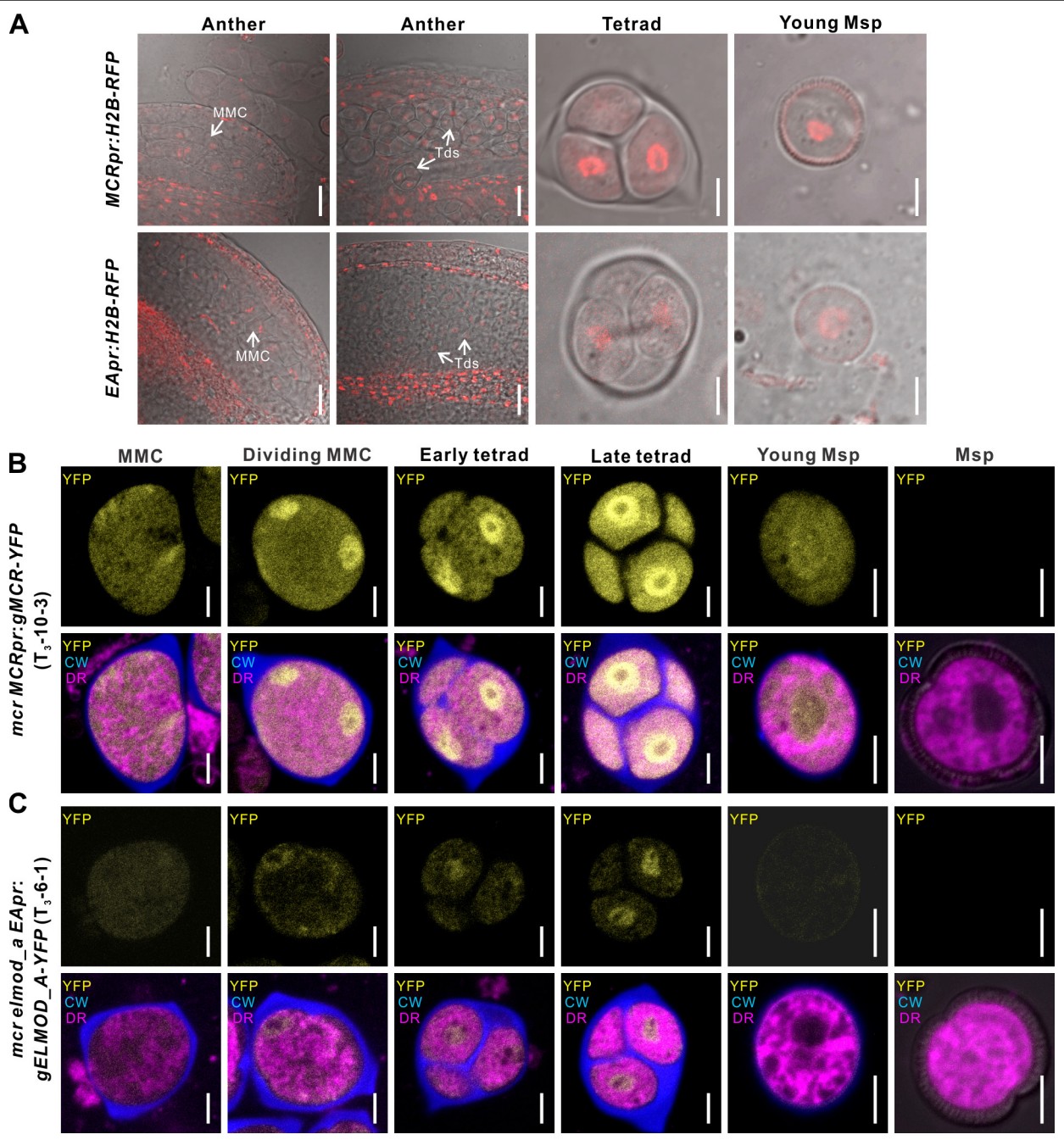

**Figure 5.** MCR and ELMOD_A do not accumulate at the aperture membrane domains. (**A**) Confocal images of wild-type anthers, tetrads, and young microspores expressing *MCRpr:H2B-RFP* (upper panels) and *EApr:H2B-RFP* (lower panels). Scale bars, 20 μm for anthers and 5 μm for tetrads and young microspores. (**B, C**) Confocal images of cells in the developing pollen lineage from *mcr MCRpr:gMCR-YFP* (**B**) and *mcr elmod_a EApr:gELMOD_A-YFP* (**C**) plants. Upper panels: YFP signal. Lower panels: merged signal from YFP (yellow), Calcofluor White (blue, callose wall), and CellMask Deep Red (magenta, membranous structures). Scale bars, 5 μm. Identical staining and color scheme are used for similar images of tetrads in other figures. CW: Calcofluor White; DR: CellMask Deep Red; MMC: microspore mother cell; Msp: microspore; Td: tetrad.

compared to the numbers in $T_1$, consistent with the transgene dosage increasing in later generations due to attaining homozygosity.

To further test the notion that aperture number depends on the *MCR/ELMOD_A* gene dosage/levels of expression, we modulated the dosage of *MCR*, starting with a defined transgene. We crossed a homozygous *mcr MCRpr:gMCR-YFP* plant from line 7-2, commonly producing >6 apertures

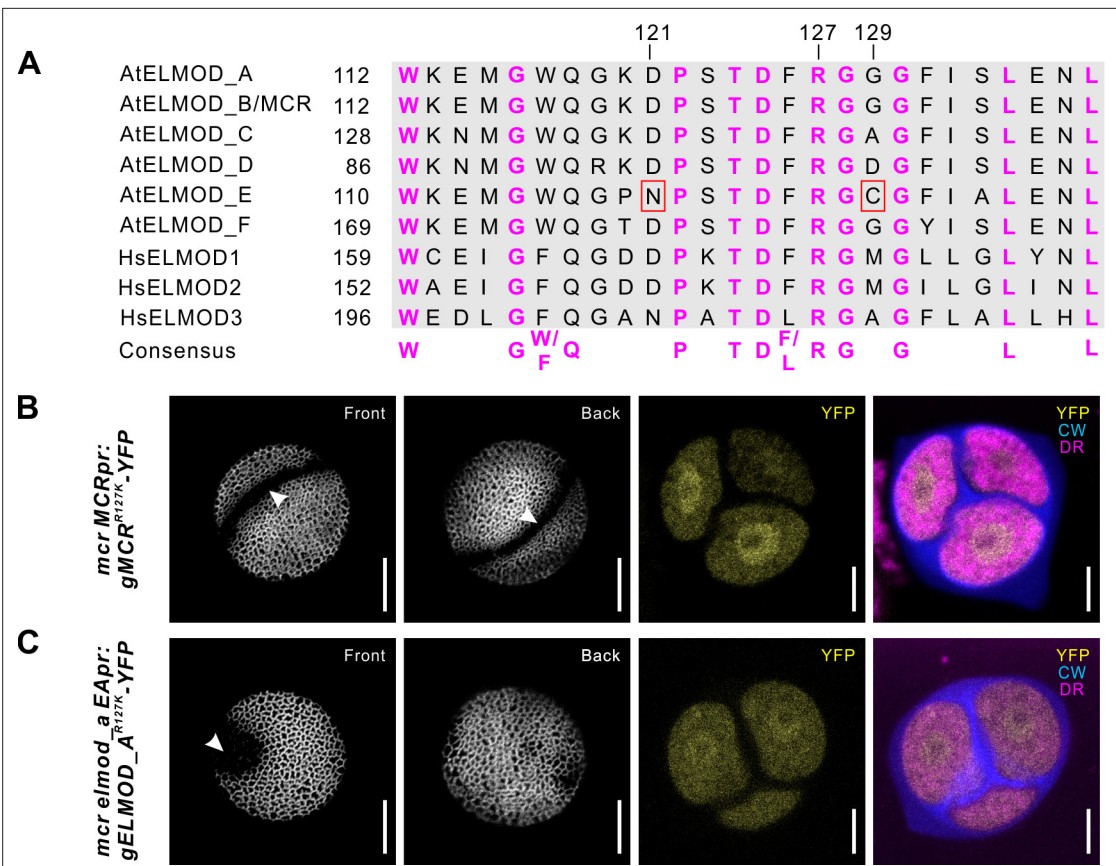

**Figure 6.** The R127 residue of MCR and ELMOD_A is essential for aperture formation. (**A**) Sequence alignment of the conserved GAP regions from six *Arabidopsis* (At) and three human (Hs) ELMOD proteins, along with the consensus sequence. Invariant Arg residue (R127) and two other important residues (121 and 129) are indicated. N121 and C129, essential for AtELMOD_E function, are indicated by red squares. (**B, C**) Confocal images of pollen grains and tetrads from *mcr* and *mcr elmod_a* expressing, respectively, *MCRpr:MCR*$^{R127K}$*-YFP* (**B**) and *EApr:ELMOD_A*$^{R127K}$*-YFP* (**C**). Apertures are indicated with arrowheads. Scale bars, 10 μm for pollen and 5 μm for tetrads.

(*Figure 7C*), with (1) *mcr* and (2) wild type. In the resulting transgenic $F_1$ progeny of the first cross, *MCR* should be expressed from one source – a single copy of the transgene. In the $F_1$ progeny of the second cross, it should be expressed from two sources – one copy of the transgene plus one of the endogenous gene. In the pollen of these $F_1$ plants, the number of apertures correlated with the number of functional copies of *MCR*: pollen of *gMCR-YFP/- mcr/mcr* produced on average 4.68 ± 1.08 apertures compared to 5.85 ± 1.52 apertures in *gMCR-YFP/- mcr/+* (*Figure 7F*, *Figure 7—figure supplement 1B*). We further assessed aperture phenotypes in the progeny of these plants that had a homozygous transgene and either zero or two copies of endogenous *MCR*. Both genotypes with the homozygous transgene produced many more apertures compared to plants with the hemizygous transgene, but they also differed significantly from each other, with the number of apertures correlating with the presence of endogenous *MCR* (8.08 ± 1.57 in *MCR-YFP/MCR-YFP mcr/mcr* vs. 9.34 ± 1.50 in *MCR-YFP/MCR-YFP +/+*) (*Figure 7F*, *Figure 7—figure supplement 1B*). These results indicate that the process of aperture domain specification is highly sensitive to the levels of MCR and ELMOD_A in developing microspores.

## The ELMOD family in angiosperms has four distinct protein clades, with most species containing two A/B type proteins

To examine the evolutionary history of the plant ELMOD family, we retrieved 561 ELMOD sequences belonging to 178 species across the plant kingdom and used them for a detailed phylogenetic analysis. ELMOD proteins are widespread in plants, suggesting that they perform important functions (*Figure 8A*).

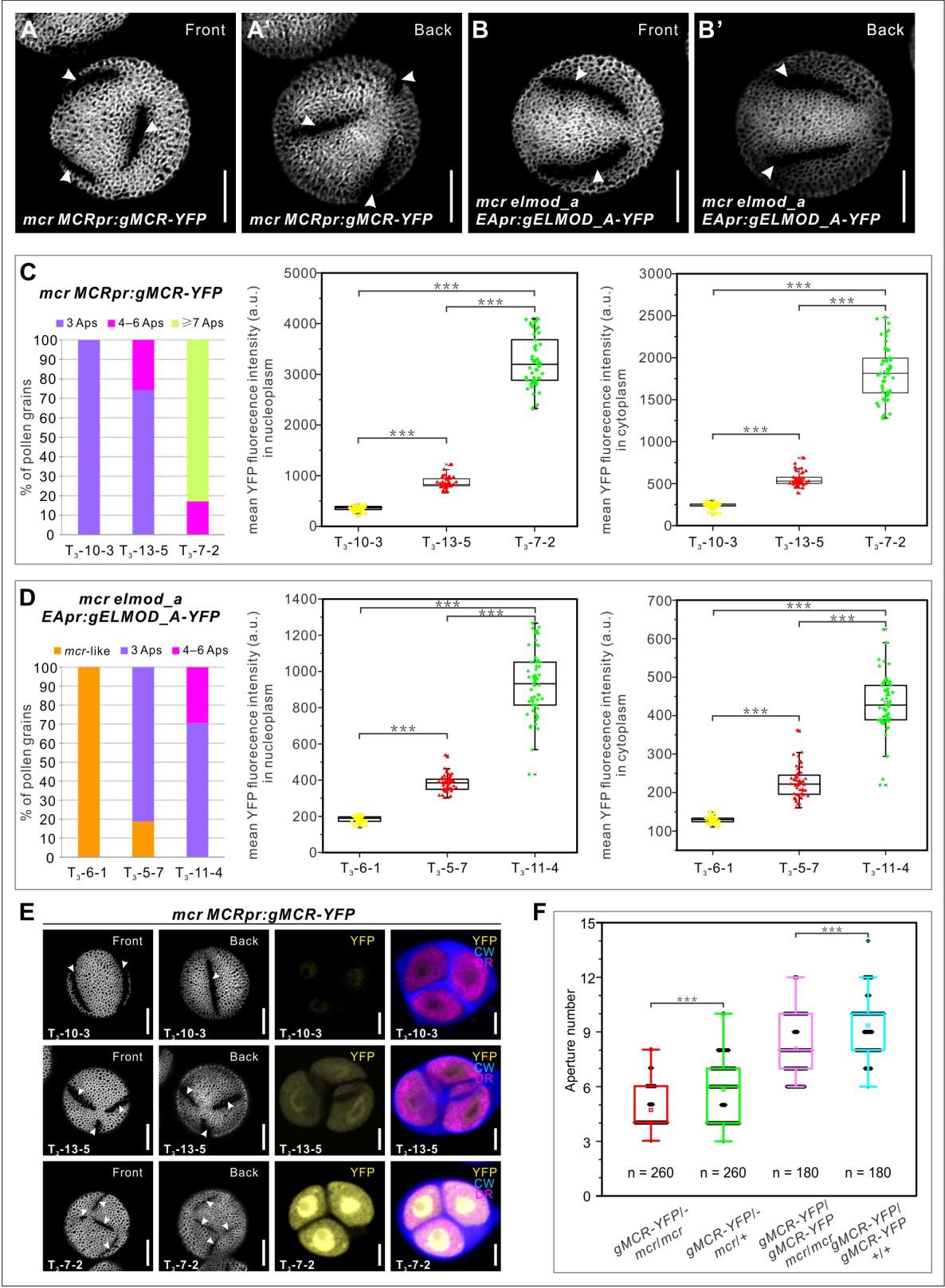

**Figure 7.** Aperture number is highly sensitive to the levels of MCR and ELMOD_A. (**A–B'**) Pollen grains from the *mcr MCRpr:gMCR-YFP* and *mcr elmod_a EApr:gELMOD_A-YFP* transgenic lines, respectively, with six and four apertures. (**C, D**) Quantification of aperture number and mean YFP signal in three homozygous lines of *mcr MCRpr:gMCR-YFP* (**C**) and *mcr elmod_a EApr:gELMOD_A-YFP* (**D**). Stacked bars show the percentage of pollen grains (from ≥3 individual plants) with indicated number of apertures. Boxplots show mean YFP signal in the microspore nucleoplasm and cytoplasm. a. u.:

*Figure 7 continued on next page*

*Figure 7 continued*

arbitrary units. (**E**) Representative images of pollen grains and tetrads corresponding to data in (**C**). (**F**) Boxplots showing aperture number depends on the number of functional copies of *MCR*. Number of analyzed pollen grains (from ≥3 individual plants) is indicated. For all boxplots, boxes represent the first and third quartiles, central lines depict the median, small squares in the boxes indicate the mean values, and small shapes show individual samples. Whiskers extend to minimum and maximum values. ***p<0.001 (two-tailed Student's t-test). Apertures are indicated with arrowheads. Scale bars, 10 µm for pollen and 5 µm for tetrads.

The online version of this article includes the following figure supplement(s) for figure 7:

**Figure supplement 1.** Representative aperture phenotypes and tetrads related to *Figure 7*.

Green algae as well as non-vascular land plants (liverworts, mosses, and hornworts) typically have a single ELMOD protein, but an ancestor of lycophytes and ferns had a gene duplication (*Figure 8A and B*). Beginning with gymnosperms, the ELMOD family expanded and diversified, with distinct protein groups clustering with the A/B/C clade, the E clade, and the F clade (*Arabidopsis* proteins were used as landmarks in naming the clades). In early angiosperms, ELMOD proteins separated into four well-supported clades: A/B, C, E, and F (*Figure 8A and B*, *Figure 8—figure supplement 1*). The split within the aperture factor-containing A/B clade into the separate ELMOD_A and ELMOD_B (MCR) lineages happened late – in the common ancestor of the Brassicaceae family (*Figure 8A*, *Figure 8—figure supplement 1*). Yet, in many other angiosperm species, including magnoliids, monocots, basal eudicots, and multiple asterids and rosids, the A/B clade also contains at least two proteins (*Figure 8—figure supplement 1*). This shows that independent duplications in this lineage happened multiple times, suggesting the existence of strong evolutionary pressure to maintain more than one gene of the A/B type.

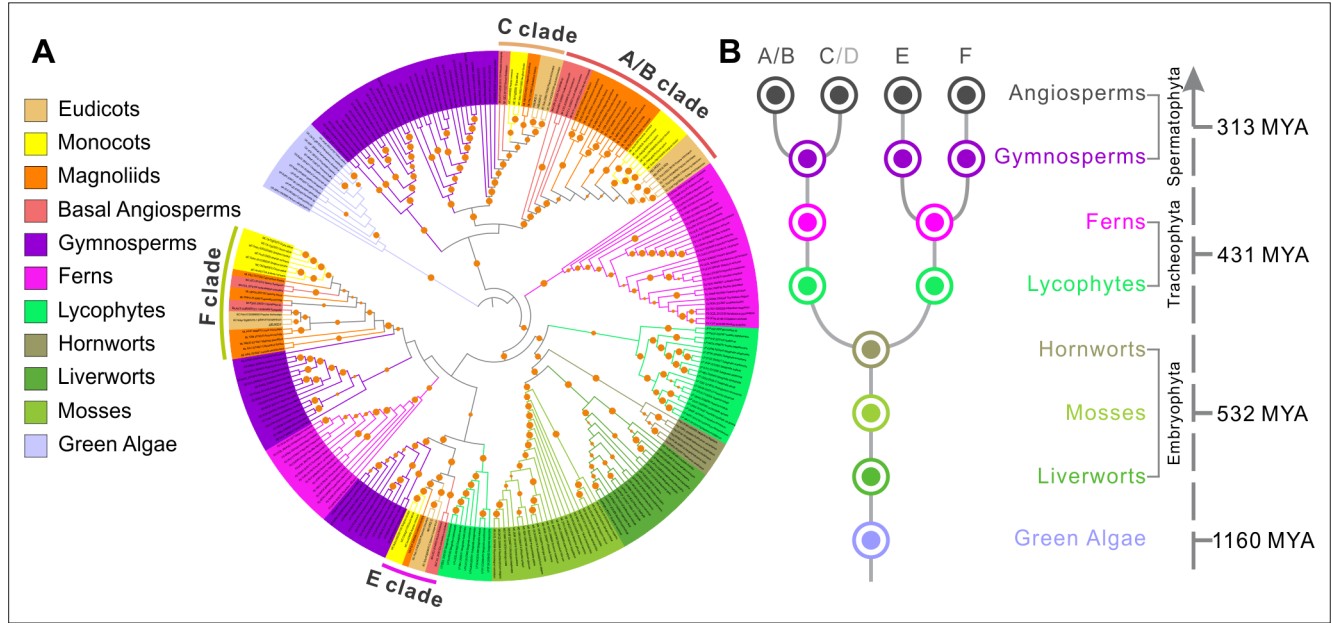

**Figure 8.** ELMOD proteins exist across the plant kingdom. (**A**) Maximum likelihood phylogenetic tree of ELMOD proteins across the plant kingdom. The four clades of angiosperm ELMODs are indicated. Orange circles: bootstrap values of 70–100%. (**B**) Inferred evolutionary history of the *ELMOD* gene family. Dots: inferred ancestral gene number in different plant groups; letters on top: ELMOD clades named after the corresponding *Arabidopsis* proteins; gray D indicates *Arabidopsis ELMOD_D* is likely a pseudogene; numbers on the right: estimated time of divergence in millions of years (MYA) calculated using the TimeTree database.

The online version of this article includes the following figure supplement(s) for figure 8:

**Figure supplement 1.** Angiosperm ELMOD proteins cluster into four clades.

## Phylogenetic analysis of the ELMOD family reveals the importance of positions 165 and 129 and suggests ELMOD_D is likely a pseudogene

The extensive number of the retrieved ELMOD sequences allowed us to evaluate conservation of the residues disrupted in MCR by the *mcr-1* and *mcr-2* mutations. Pro165, converted into Ser in *mcr-1* (*Figure 3A*, *Figure 3—figure supplement 1*), was present in each of the 553 ELMOD sequences containing this region, suggesting a critical role in protein function. This Pro belongs to the highly conserved WEY**P**FAVAG motif (*Figure 3—figure supplement 1*) found in all six *Arabidopsis* ELMODs, as well as in the majority of ELMODs from other plants, including green algae.

The appearance of Asp at position 129 in *mcr-2* (*Figure 3A*, *Figure 3—figure supplement 1*) affects a site within the putative GAP region, neighboring the critical catalytic Arg127. This change is also highly unusual from the evolutionary perspective. Except for one likely pseudogene (see below), none of the other 560 ELMOD sequences from across the plant kingdom has an Asp at that site, consistent with the notion that an Asp at this position is not tolerated by natural selection and could be detrimental for all plant ELMOD proteins.

Our analysis of residues occupying position 129 in the GAP region across the angiosperm ELMOD proteins led to an interesting discovery. In the 365 analyzed angiosperm sequences, this site is occupied by only three amino acids: Cys, Gly, or Ala. (Earlier diverged plants have Ala or Gly at this site.) Strikingly, we found that all proteins with Cys129 cluster with the E clade, whereas nearly all proteins with Gly129 cluster with either the A/B or the F clades, and nearly all proteins with Ala129 cluster with the C clade. (Only six exceptions were found among the 365 sequences: in five cases, proteins containing Ala129 clustered with the A/B or the F clades, and in one case, a protein with Gly129 clustered with the C clade.) This suggested the intriguing possibility, tested later, that, in angiosperms, residues at position 129 are important for functional differentiation of the ELMOD proteins.

Besides *mcr-2*, the only protein with Asp at position 129 is the *Arabidopsis* ELMOD_D. However, it has several other features that suggest it is likely a pseudogene. At 213 amino acids, ELMOD_D is markedly shorter than the other five *Arabidopsis* ELMODs (265–323 aa-long): it misses stretches of 52 aa upstream of the GAP region, 4 aa in the vicinity of the GAP region, and 22 aa at the very C-terminus of the protein (*Figure 3—figure supplement 1*). It also has major substitutions unique to this protein within or near its GAP region, which change the conserved Gly119 and Leu138 residues into Arg (the numbering within the GAP region is based on the MCR and ELMOD_A sequences; *Figure 3—figure supplement 1*). ELMOD_D clusters with the C clade and is most closely related to the *Arabidopsis* ELMOD_C, indicating that it is a product of a very recent duplication (*Figure 8—figure supplement 1*). While some plants have more than one protein in the C clade, most others, including close relatives of *Arabidopsis*, have just a single C protein (*Figure 8—figure supplement 1*), suggesting that a single C-type activity is sufficient for most species. These findings, combined with the extremely low levels of *ELMOD_D* expression (*Figure 3E*), support the hypothesis that this member of the *Arabidopsis* ELMOD family is likely non-functional.

## ELMOD_E can influence aperture formation and produce a novel aperture pattern

To test if other ELMOD family members, besides MCR and ELMOD_A, might be involved in aperture formation, we examined the phenotypes of their single mutants and double mutants with *mcr* (*Figure 9—figure supplement 1*). All single mutants displayed normal aperture patterns (*Figure 9—figure supplement 1B–F'*),while the double mutants exhibited *mcr* phenotypes (*Figure 9—figure supplement 1G–J'*), indicating that, unlike *ELMOD_A*, the other four *ELMOD* genes do not interact synergistically with *MCR* in aperture formation.

We also assessed the ability of these genes to rescue the *mcr* aperture phenotype by expressing them, tagged with *YFP*, from the *MCR* regulatory regions. *ELMOD_C* showed some limited ability to restore three apertures in *mcr* (*Figure 9—figure supplement 2A–D'*), while *ELMOD_D* and *ELMOD_F* were unable to do it (6/6 and 22/22 T$_1$ plants; *Figure 9—figure supplement 2E–F'*). Based on the YFP signal, ELMOD_F-YFP was expressed at a level comparable with that of the MCR-YFP and ELMOD_A-YFP transgenes (*Figure 9—figure supplement 2G and G'*). ELMOD_D-YFP, however, was undetectable (*Figure 9—figure supplement 2H and H'*), consistent with the hypothesis that *ELMOD_D* is a pseudogene.

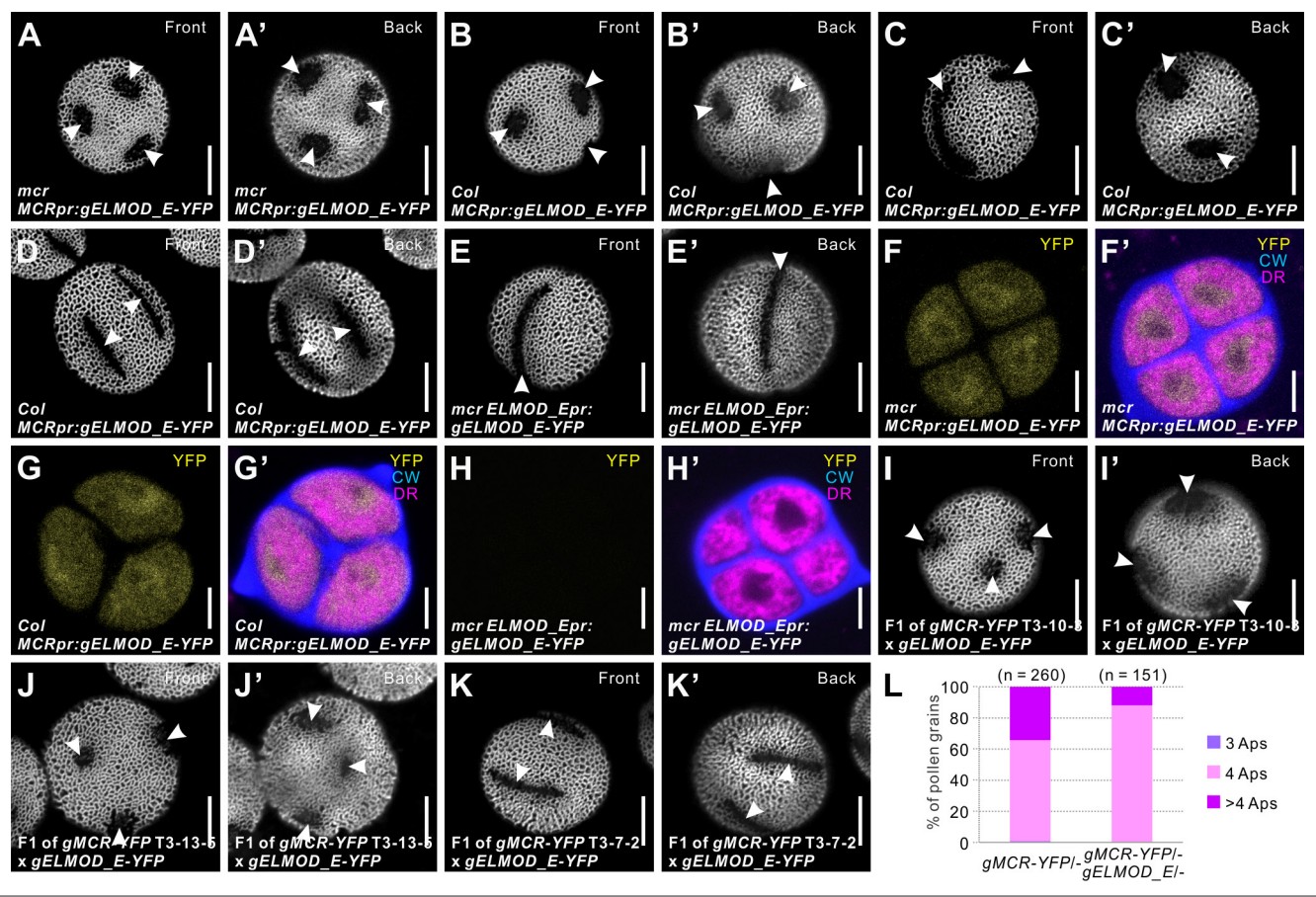

**Figure 9.** *Arabidopsis* ELMOD_E can affect aperture patterns. (**A–D′**) Pollen grains from *mcr* (**A, A′**) and Col-0 (**B–D′**) plants expressing *MCRpr:gELMOD_E-YFP*. (**E, E′**) Pollen grain from *mcr* plants expressing *ELMOD_Epr:gELMOD_E-YFP*. (**F–H′**) Confocal images of tetrads expressing *MCRpr:gELMOD_E-YFP* and *ELMOD_Epr:gELMOD_E-YFP*. Adjacent panels show YFP signal (α) and merged signal (α′) from YFP, Calcofluor White (CW), and CellMask Deep Red (DR). (**I–K′**) Pollen grains from the F₁ plants produced by crossing *mcr MCRpr:gELMOD_E-YFP* with three T₃ lines of *mcr MCRpr:gMCR-YFP* (with single homozygous insertions of the *MCR-YFP* transgene, expressed, respectively, at low, medium, and high levels). (**L**) Percentage of pollen grains with indicated number of apertures in the pollen populations from F₁ progeny of the *mcr MCRpr:gMCR-YFP* T₃-7-2 line crossed with *mcr* or with *mcr MCRpr:gELMOD_E-YFP*. Number of analyzed pollen grains (from at least two individual plants) is indicated. Apertures are indicated with arrowheads. Scale bars, 10 µm for pollen and 5 µm for tetrads.

The online version of this article includes the following figure supplement(s) for figure 9:

**Figure supplement 1.** Disruptions of *Arabidopsis ELMOD_C, ELMOD_D, ELMOD_E,* and *ELMOD_F* do not affect aperture patterns.

**Figure supplement 2.** ELMOD_C, but not ELMOD_D and ELMOD_F, can partially substitute for MCR in aperture formation.

Most interestingly, the expression of *ELMOD_E* in *mcr* led to a neomorphic phenotype: instead of narrow longitudinal furrows, pollen of all seven T₁ plants developed multiple short, round apertures (*Figure 9A and A′*). To see if this effect was limited to the *mcr* background, we transformed the *MCRpr:ELMOD_E-YFP* construct into wild-type Col-0 plants. The T₁ plants showed a range of aperture phenotypes (*Figure 9B–D′*), yet multiple round apertures were commonly present, suggesting that *ELMOD_E* exerts a dominant negative effect when misexpressed in developing microspores.

We then tested whether *ELMOD_E* would have the same effect on aperture patterns when expressed from its own promoter. However, none of the 12 T₁ transgenic *mcr* plants expressing the *ELMOD_Epr:ELMOD_E-YFP* construct had any changes in the *mcr* aperture phenotype (*Figure 9E and E′*). Analysis of the YFP signal showed that this gene is expressed in tetrad-stage microspores at much lower levels from its own promoter than from the *MCR* promoter (*Figure 9F–H′*). Thus, while ELMOD_E can influence aperture patterns, it is likely not normally involved in this process in *Arabidopsis*.

To test if differences in the MCR levels could impact the ability of transgenic ELMOD_E to produce round apertures, we crossed a *mcr MCRpr:gELMOD_E-YFP* line with the above described *mcr MCRpr:gMCR-YFP* lines 10-3, 13-5, and 7-2 that express MCR, respectively, at low, medium, and high levels (*Figure 7C*). In the $F_1$ progeny of crosses with low and medium expressors, 10-3 and 13-5, pollen still produced round apertures (*Figure 9I–J'*). Yet, in the $F_1$ progeny of the cross with the high expressor 7-2, furrow aperture morphology was restored (*Figure 9K and K'*), although these plants produced less pollen with a high number of furrows (>4) compared to the original 7-2 plants (*Figure 9L*). Together, these data support the idea that high level of MCR can counteract the neomorphic activity of ELMOD_E and suggest that MCR and ELMOD_E may compete for the same interactors.

### Residues 121 and 129 in the GAP region are important for the MCR- and ELMOD_E-specific functions in aperture formation

The different aperture phenotypes of *mcr MCRpr:gMCR-YFP* and *mcr MCRpr:gELMOD_E-YFP* lines gave us an opportunity to test the hypothesis that residues at position 129 are important for functional differentiation of ELMODs from different clades. For E-clade proteins, we also noticed that

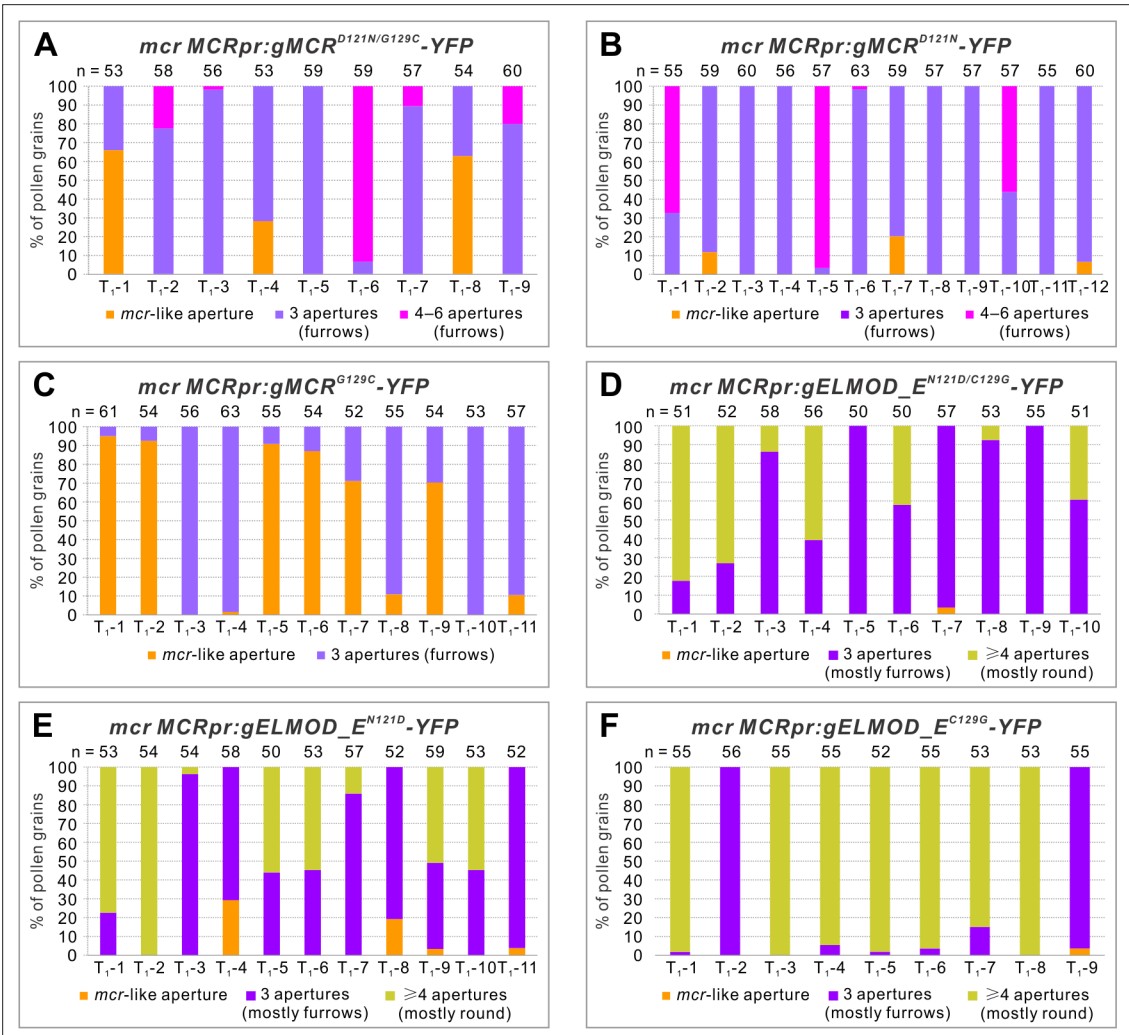

**Figure 10.** Residues 121 and 129 in the GAP region are important for MCR- and ELMOD_E-specific functions in aperture formation. Percentage of pollen grains with indicated number of apertures in the pollen populations from independent $T_1$ *mcr* plants expressing variants of *MCRpr:gMCR-YFP* (**A–C**) or *MCRpr:ELMOD_A-YFP* (**D–F**) with residues 121 and/or 129 mutated. Number of analyzed pollen grains is indicated above the bars.

The online version of this article includes the following figure supplement(s) for figure 10:

**Figure supplement 1.** Representative aperture phenotypes observed in $T_1$ plants related to *Figure 10*.

Cys129 was always found together with Asn121. These residues are unique to this clade: 100% of the retrieved E-clade sequences (n = 69) have Asn121/Cys129 vs. 0% of sequences from the other clades (n = 297). Thus, this combination could be important for the E-clade functions. In the other three clades, position 121 is always occupied by Asp.

To investigate the importance of sites 121 and 129 for MCR and ELMOD_E functions, we created six constructs in which one or both residues at these positions were replaced with the residues typical of the other clade and expressed them in the *mcr* mutant. The MCR proteins carrying the E-specific residues at both positions (MCR$^{D121N/G129C}$) or at the position 121 (MCR$^{D121N}$) still retained most of the MCR function, with most T$_1$ plants producing three or more furrow apertures in most of their pollen grains (7/9 and 12/12 T$_1$ plants, respectively; *Figure 10A and B*, *Figure 10—figure supplement 1A*). However, when the E-specific residue was present only at position 129 (MCR$^{G129C}$), MCR protein became less active, with only 5 out of 11 T$_1$ plants producing three furrow apertures in all or most of their pollen (*Figure 10C*). In the rest of these T$_1$ plants, the *mcr* phenotype was not rescued or was rescued poorly, with <30% of pollen grains forming three apertures.

Experiments with the ELMOD_E proteins carrying the MCR residues at positions 121 and 129 confirmed the importance of Asn121 and Cys129 for the ELMOD_E neomorphic activity. In the case where both residues were replaced with the MCR residues (ELMOD_E$^{N121D/C129G}$), ELMOD_E largely lost its ability to create round apertures and instead often restored three furrow-like apertures, thus acting like MCR (*Figure 10D*, *Figure 10—figure supplement 1B*). In the cases when only one residue was changed (ELMOD_E$^{N121D}$ and ELMOD_E$^{C129G}$), the mutant ELMOD_E proteins were still often able to produce multiple round apertures, although three normal furrows or a mixture of furrows and round apertures were also produced, suggesting that the single mutations reduced the ELMOD_E activity, but did not eliminate it entirely (*Figure 10E and F*, *Figure 10—figure supplement 1B*). While the number of apertures produced in these experiments varied and some pollen grains had both furrows and round apertures (*Figure 10—figure supplement 1B*), in general, there was a strong correlation between the number of apertures per pollen and their shape. Analysis of 207 pollen grains from across the ELMOD_E$^{N121D/C129G}$ transgenic lines showed that when pollen had three apertures, they were mostly represented by furrows (95%, n = 468 apertures), whereas in pollen with four or more apertures, apertures were mostly round (84%, n = 239 apertures; *Figure 10—figure supplement 1C*). Similar trends were also observed in ELMOD_E$^{N121D}$ and ELMOD_E$^{C129G}$ transgenic lines (*Figure 10—figure supplement 1C*).

Taken together, these results show that residues at positions 121 and 129 in the GAP region provide important contributions to the specific functions of MCR and ELMOD_E. Yet they are less critical for MCR, in accord with the fact that Asp121 and Gly129 are not unique to the A/B clade. In the case of ELMOD_E, the E-clade-specific combination of Asn121/Cys129 appears to be essential for its distinct activity. When both residues undergo MCR-like changes, ELMOD_E loses its neomorphic activity, instead becoming capable of carrying out the MCR role in aperture formation.

## Discussion

How developing pollen grains create beautiful and diverse geometrical aperture patterns has been a long-standing problem in plant biology (*Fischer, 1889*; *Ressayre et al., 2002*; *Wodehouse, 1935*). In this study, we uncovered the first set of molecular factors, belonging to the ELMOD protein family, that have a clear ability to regulate the number, positions, and morphology of aperture domains. MCR and its close paralog ELMOD_A act as (somewhat) redundant positive regulators of furrow aperture formation in *Arabidopsis*.

Our genetic analysis places MCR and ELMOD_A at the beginning of the aperture formation pathway, upstream of the previously discovered aperture factors D6PKL3, INP1, and, likely, INP2, the recently identified partner of INP1. Previous studies showed that INP1 and INP2 act as the executors of the aperture formation program, absolutely essential for aperture development but not able on their own to influence the number and positions of aperture domains (*Dobritsa et al., 2018*; *Lee et al., 2021*; *Li et al., 2018*; *Reeder et al., 2016*). D6PKL3 was proposed to act upstream of INP1, defining the features of aperture domains, yet it also largely lacks the ability to initiate completely new domains (*Lee et al., 2018*; *Zhou and Dobritsa, 2019*). In *mcr* microspores, D6PKL3 and INP1 re-localize to the ring-shaped aperture domains (*Figures 1G and 2A*), indicating that they become

attracted to the newly specified aperture domains and the ELMOD proteins act as patterning factors, contributing to symmetry breaking and selection of sites for aperture domains.

Our data demonstrate that the aperture domains forming in each microspore are highly sensitive to the ELMOD_A/MCR protein dosage (*Figure 4C–G'*, *Figure 7*, *Figure 7—figure supplement 1*). Increased dosage leads to a higher number of apertures, while decreased dosage results in fewer, and the reducing effect of the loss in *MCR* activity on aperture number is maintained across different levels of ploidy and post-meiotic arrangements of microspores (*Figure 1I*, *Figure 1—figure supplement 2*), which were previously shown to be important factors in aperture patterning (*Reeder et al., 2016*). Thus, modulation of ELMOD protein levels might offer a mechanism for creating different aperture patterns in different species. Interestingly, within the genus *Pedicularis*, some species display the *mcr*-like ring-shaped apertures, while others produce three apertures (*Wang et al., 2009*; *Wang et al., 2017*). Our findings suggest that such variations in aperture patterns could conceivably be due to variations in ELMOD proteins or their effectors or regulators. Importantly, while great diversity of pollen aperture numbers is found across plant species, within a species, this trait tends to be very robust. For example, in wild-type *Arabidopsis*, the number of apertures rarely deviates from three (*Reeder et al., 2016*). Our results, therefore, imply that, normally, levels of MCR and ELMOD_A are very tightly controlled, and there must exist mechanisms to ensure this control.

The discovery that *ELMOD_E* can also influence aperture patterns in *Arabidopsis* and create multiple round apertures instead of three furrows (*Figure 9A–B'*) suggests that the regulation of *ELMOD_E* might also contribute to the diversity of aperture patterns in nature. In *Arabidopsis*, *ELMOD_E* does not seem to be usually involved in aperture formation. Yet, when misexpressed from the *MCR* regulatory regions, it interferes with MCR and ELMOD_A activity (*Figure 9I–L*), resulting in the formation of new aperture domains.

ELMODs are ancient proteins, predicted to have been present in the last common ancestor of all eukaryotes (*East et al., 2012*). In animals, these proteins act as non-canonical GAPs, regulating activities of both Arf and Arl GTPases (*Bowzard et al., 2007*; *Ivanova et al., 2014*; *Turn et al., 2020*). Arf GTPases are commonly associated with the recruitment of vesicle coat proteins to different membrane compartments to initiate vesicle budding and trafficking, while the roles of the related Arl proteins are less understood and likely more diverse (*Sztul et al., 2019*). Although the function of ELMOD proteins in plants is unknown, their presence in green algae and other basal plants lineages suggests that they have been playing important roles in plant cells since their inception. Our phylogenetic analysis indicates that this family in plants is monophyletic, and the genes have duplicated and diversified over the course of plant evolution.

The angiosperm ELMOD family has four distinct clades (*Figure 8A and B*, *Figure 8—figure supplement 1*). In many species, the A/B clade, containing MCR and ELMOD_A, has two or more proteins due to independent duplications that occurred multiple times in evolution. This suggests that species might be under a selective pressure to keep more than one A/B type protein, implying that the processes in which these proteins are involved (e.g., aperture formation) benefit from genetic redundancy and, thus, are highly important.

Further studies will be required to establish the biochemical role of plant ELMOD proteins. Like their animal counterparts, plant ELMODs may be involved in regulation of Arf/Arl activities. *Arabidopsis* has 19 ARFs and ARLs, which, with few exceptions, mostly remain uncharacterized (*Delgadillo et al., 2020*; *Gebbie et al., 2005*; *McElver et al., 2000* ; *Singh et al., 2018*; *Vernoud et al., 2003*; *Xu and Scheres, 2005*). The roles attributed to members of this family – for example, in secretion, endocytosis, activation of phosphatidyl inositol kinases, and actin polymerization (*Singh and Jürgens, 2018*; *Sztul et al., 2019*) – are all potentially fitting with the formation of distinct aperture domains. The protein region proposed to be the GAP region in mammalian ELMODs (*East et al., 2012*) is conserved in plant proteins, and the invariant Arg residue believed to be catalytic in mammalian ELMODs is also necessary for function in MCR and ELMOD_A (*Figure 6B and C*). Interestingly, some positions within the conserved GAP region show strict residue specificity in different clades, suggesting that they could be important for functional diversity of these proteins. Consistent with this, we found the combination of Asn121/Cys129 to be key for the ELMOD_E neomorphic aperture-forming activity (*Figure 10*).

Alternatively, plant ELMODs could have evolved functions different from their animal counterparts and regulate targets other than ARFs/ARLs. Interestingly, the only study done so far on an ELMOD

protein in plants (**Hoefle and Hückelhoven, 2014**) pulled out the barley homolog of ELMOD_C in a yeast two-hybrid screen as an interactor of a ROP GAP, a GAP for a different class of small GTPases, Rho-of-plants (ROPs). Rho GTPases (including ROPs) are well-known regulators of cell polarity and domain formation (**Feiguelman et al., 2018**; **Yang and Lavagi, 2012**), so their involvement in aperture formation cannot be excluded.

In summary, we presented critical players in the process of patterning the pollen surface. These players belong to the ELMOD protein family, which, while undoubtedly important, has not yet been characterized in plants. Future studies should focus on identifying the interactors of the ELMOD proteins and on understanding the mechanisms through which they specify positions and shape of aperture domains without noticeably accumulating at these regions.

# Materials and methods

## Key resources table

| Reagent type (species) or resource | Designation | Source or reference | Identifiers | Additional information |
|---|---|---|---|---|
| Gene (*Arabidopsis thaliana*) | ELMOD_A | https://www.arabidopsis.org/ | AT3G60260 | N/A |
| Gene (*Arabidopsis thaliana*) | MCR/ELMOD_B | https://www.arabidopsis.org/ | AT2G44770 | N/A |
| Gene (*Arabidopsis thaliana*) | ELMOD_C | https://www.arabidopsis.org/ | AT1G67400 | N/A |
| Gene (*Arabidopsis thaliana*) | ELMOD_D | https://www.arabidopsis.org/ | AT3G43400 | N/A |
| Gene (*Arabidopsis thaliana*) | ELMOD_E | https://www.arabidopsis.org/ | AT1G03620 | N/A |
| Gene (*Arabidopsis thaliana*) | ELMOD_F | https://www.arabidopsis.org/ | AT3G03610 | N/A |
| Strain, strain background (*Agrobacterium tumefaciens*) | GV3101 | Widely distributed | N/A | Competent cells |
| Genetic reagent (*Arabidopsis thaliana*) | mcr-1 | This study, EMS mutagenesis | N/A | See **Supplementary file 1** for all other genetic reagents |
| Chemical compound, drug | Auramine O | Thermo Fisher Scientific | A96825 | N/A |
| Chemical compound, drug | Vectashield antifade solution | Vector Labs | H-1000-10 | N/A |
| Chemical compound, drug | Calcofluor White | PhytoTechnology Laboratories | C1933 | N/A |
| Chemical compound, drug | CellMask Deep Red | Thermo Fisher Scientific | C10046 | N/A |
| Software, algorithm | NIS Elements v.4.20 | Nikon Microscopy | N/A | N/A |
| Software, algorithm | MAFFT version 7 | **Katoh and Standley, 2013**; **Katoh et al., 2002** | https://mafft.cbrc.jp/alignment/software/ | N/A |
| Software, algorithm | TrimAl | **Capella-Gutiérrez et al., 2009** | https://vicfero.github.io/trimal/ | N/A |
| Software, algorithm | IQ-TREE | **Nguyen et al., 2015** | http://www.iqtree.org/ | N/A |
| Software, algorithm | Origin version 2018 | OriginLab | https://www.originlab.com/ | N/A |

## Plant materials and growth conditions

*Arabidopsis thaliana* genotypes used in this study were either in Columbia (Col) or Landsberg *erecta* (L*er*) background. Pollen from wild-type Col- and L*er* has indistinguishable aperture pheno-types. The following genotypes were also used: *mcr-1*, *mcr-2*, *mcr-3*, *mcr-4*, *mcr-5* (CS853233), *mcr-6* (SALK_205528C ), *mcr-7* (SALK_203827C ), *elmod_c* (SALK_076565), *elmod_d* (SALK_031512), *elmod_e* (SALK_082496), *elmod_f* (SALK_010379), *inp1-1* (**Dobritsa and Coerper, 2012**), *inp2-1* (**Lee et al., 2021**), *d6pkl3-2* (**Lee et al., 2018**), *inp1-1 DMC1pr:INP1-YFP* (**Dobritsa et al., 2018**), *d6pkl3-2 D6PKL3pr:D6PKL3-YFP* (**Lee et al., 2018**), *tes* (SALK _113909), *MiMe* (**d'Erfurth et al., 2009**), and *cenh3-1 GFP-tailswap* (CS66982). *mcr-1* through *mcr-4* mutants were discovered in a forward genetic screen performed on an EMS-mutagenized L*er* population (**Plourde et al., 2019**). *mcr-5* through *mcr-7* mutants and *elmod_c* through *elmod_f* mutants were ordered from the Arabidopsis Biological

Resource Center (ABRC). Plants were grown at 20–22°C with the 16 hr light/8 hr dark cycle in growth chambers or in a greenhouse at the Biotechnology support facility at OSU.

To generate the 2n *mcr tes* plants, *mcr-1* mutant was crossed with heterozygous *tes*, double heterozygotes were recovered in F$_1$ by genotyping (primers listed in *Supplementary file 1*), and double homozygotes were identified in F$_2$ population. The generation of haploid *mcr MiMe* plants was similar to the procedure previously described (*Reeder et al., 2016*). In brief, *mcr-1* mutant was first crossed with plants that were triple heterozygotes for *atrec8-3*, *osd1-3*, and *atspo11-1-3* (*MiMe* heterozygotes), then the quadruple heterozygotes were identified among the F$_1$ progeny by genotyping and crossed as males with *cenh3-1 GFP-tailswap* homozygous plants that were used as haploidy inducers (*Ravi and Chan, 2010*). 1n F$_1$ progeny of this cross were identified by their distinctive morphology as described (*Ravi and Chan, 2010*; *Reeder et al., 2016*), and the triple 1n *MiMe* and quadruple 1n *mcr MiMe* mutants were identified by genotyping (primers listed in *Supplementary file 1*). Unlike other 1n genotypes generated by this cross, which were sterile, the 1n plants with *MiMe* mutations were fertile and produced 1n pollen via mitosis-like division and dyad formation.

Mapping of the *MCR* locus *mcr-1* mutant with L*er* background was crossed with Col-0, and individual F$_2$ plants were screened under a dissecting microscope for the presence of the distinctive angular mutant phenotype in their dry pollen. In total, 369 plants with mutant phenotype were selected, and their genomic DNA was isolated. To map the *MCR* locus, we first conducted bulked segregant analysis, followed by the map-based positional cloning (*Lukowitz et al., 2000*). The insertion-deletion (InDel) molecular markers were developed based on the combined information from the 1001 Genomes Project database (*1001 Genomes Consortium, 2016*) and the *Arabidopsis* Mapping Platform (*Hou et al., 2010*). The *MCR* locus was mapped to a 77 kb region between markers 2–18.39 Mb (18,395,427 bp) and 2–18.47 Mb (18,472,092 bp) on chromosome 2. Molecular markers used for mapping are listed in *Supplementary file 1*. Out of the 25 genes located in this interval, we sequenced 11 genes, prioritized based on their predicted expression patterns and gene ontology, and found that one of them, *At2g44770*, contained a missense mutation. Sequencing of the other three non-complementing EMS alleles identified in the forward genetic screen (*mcr-2* to *mcr-4*) also revealed presence of mutations in *At2g44770*.

## Inactivation of *ELMOD_A* and *ELMOD_E* with CRISPR/Cas9

Two guide RNAs against target sequences at the beginning of the *ELMOD_A* and *ELMOD_E* CDS were selected with the help of the CRISPR-PLANT platform (https://www.genome.arizona.edu/crispr/ *Xie et al., 2014*) and individually cloned into the *Bsa*I site of the pHEE401E vector (*Wang et al., 2015*) as described (*Xing et al., 2014*), using, respectively, two sets of complementary primers: *elmod_a* sgRNA-F/R and *elmod_e* sgRNA-F/R (*Supplementary file 1*). The resulting constructs were separately transformed into the *Agrobacterium tumefaciens* strain GV3101, and then used to transform *Arabidopsis* Col-0 plants or *mcr-1* mutants (the latter only with the anti-*ELMOD_E* construct) using the floral-dip method (*Clough and Bent, 1998*). The T$_1$ transformants were selected on ½ strength MS plates supplemented with 1% (w/v) sucrose, 0.8% (w/v) agar, and 50 µg/mL hygromycin, their DNA was extracted, and the regions surrounding the target sequences were sequenced. For *ELMOD_A*, 5 of 25 T$_1$ plants had homozygous, biallelic, or heterozygous mutations. Sequencing the progeny of these plants demonstrated that all homozygous/biallelic mutants developed frame shifts in the *ELMOD_A* CDS after the codon 64 (by acquiring either a 1-nt insertion three nucleotides before PAM or a 1-nt deletion two nucleotides before PAM). An *elmod_a* mutant with a single A insertion, as shown in *Figure 4A*, and still carrying CRISPR/Cas9 transgene, was crossed with the *mcr-1* mutant to obtain the *mcr elmod_a* double mutant. For *ELMOD_E*, 1 out of 12 and 1 out of 20 T$_1$ plants had biallelic mutations, respectively, in Col-0 and *mcr-1* backgrounds. In T$_2$ generation, homozygous mutants with a frame shift in the CDS were identified: in *elmod_e$^{CR}$*, a 13-nt region located four nucleotides before PAM was deleted and replaced with a different 9-nt sequence; in *mcr elmod_e$^{CR}$*, a single A was inserted four nucleotides before PAM. These mutants were used to observe the aperture phenotypes.

## Generation of transgenic constructs and plant transformation

A 3076 bp fragment upstream of the start codon of *MCR* was used as the *MCR* promoter for all *MCRpr* constructs. To generate the *MCRpr:gMCR* construct, the promoter and the 2868 bp genomic fragment from the *MCR* start codon to 798 bp downstream of the stop codon were separately amplified

from Col-0 genomic DNA and cloned into *SacI*/*NcoI* sites in the pGR111 binary vector (***Dobritsa et al., 2010***) through In-Fusion cloning (Takara). An *AgeI* site was introduced in front of the *MCR* start codon for ease of subsequent cloning. For *MCRpr:MCR CDS*, the genomic fragment was replaced with the *MCR* coding sequence, which was amplified from the *MCR* cDNA construct CD257409 obtained from ABRC. For *MCRpr:gMCR-YFP* construct, the genomic fragment of *MCR* was amplified without the stop codon and cloned upstream of *YFP* into the pGR111 binary vector (***Dobritsa et al., 2010***). Additionally, a 497 bp 3′ UTR region from *MCR* was then cloned downstream of *YFP*. Since we achieved phenotypic rescue and observed strong YFP signal with this construct, we used this combination of regulatory elements in all subsequent constructs for which we wanted to achieve the *MCR*-like expression. The constructs *MCRpr:gELMOD_A/C/D/E/F-YFP* were created in a similar way.

For all *EApr* constructs, a 2163 bp fragment upstream of the start codon of *ELMOD_A* was amplified from Col-0 genomic DNA and used as the *ELMOD_A* promoter. For *EApr:gELMOD_A*, a 2833 bp fragment, which included a 296 bp region downstream of the stop codon, was subcloned into pGR111 downstream of *EApr*. A *Bam*HI site was introduced in front of the start codon for ease of subsequent cloning. For *EApr:gELMOD_A-YFP*, a 2534 bp genomic fragment (from the *ELMOD_A* start codon to immediately upstream of the stop codon) was cloned between the *EApr* and *YFP*. For *ELMOD_Epr:ELMOD_E-YFP*, a 1469 bp fragment upstream of the start codon of *ELMOD_E* was amplified from Col-0 genomic DNA and used as the *ELMOD_E* promoter to replace the *MCR* promoter in *MCRpr:gELMOD_E-YFP*.

To generate the reporter constructs *MCRpr:H2B-RFP* and *EApr:H2B-RFP*, the *H2B-RFP* fusion gene was cloned into the *Bam*HI/*Spe*I sites downstream of the respective promoters in pGR111. To create constructs with single and double nucleotide substitutions, PCR-based site-directed mutagenesis was performed with IVA mutagenesis (***García-Nafría et al., 2016***) using gMCR-pGEM-T Easy, gELMOD_A-pGEM-T Easy, and gELMOD_E-pGEM-T Easy as templates. The mutated sequences then replaced the respective wild-type sequences in *MCRpr:gMCR-YFP*-pGR111, *EApr:ELMOD_A-YFP*-pGR111, and *MCRpr:gELMOD_E-YFP*-pGR111. All primers used for creating constructs are listed in ***Supplementary file 1***. All constructs were verified by sequencing and transformed by electroporation into the *Agrobacterium* strain GV3101 together with the helper plasmid pSoup. *Agrobacterium* cultures confirmed to contain the constructs of interest were then transformed into *mcr* or *mcr elmod_a* by floral dip (*mcr elmod_a* was verified to lack the anti-ELMOD_A CRISPR/Cas9 transgene).

## Confocal microscopy and image analysis

Preparation and imaging of mature pollen grains, MMC, tetrads, and free microspores were performed as previously described (***Reeder et al., 2016***). Imaging was done on a Nikon A1+ confocal microscope with a 100× oil-immersion objective (NA = 1.4), using 1× confocal zoom for anthers, 3× zoom for pollen grains, 5× zoom for MMC and tetrads, and 5× or 8× zoom for free microspores. For imaging mature pollen grains, pollen was placed into an ~10 μL drop of auramine O working solution (0.001% ; diluted in water from the 0.1% [w/v] stock prepared in 50 mM Tris-HCl), allowed to hydrate for ~5 min, covered with a #1.5 cover slip, and sealed with nail polish. Exine was excited with a 488 nm laser and fluorescence was collected at 500–550 nm. To count aperture number, images from the front and back view of pollen grain were taken. If some apertures were present on sides of a pollen grain not directly visible by focusing on the front and on the back, then z-stacks were taken (step size = 500 nm) and 3D images were reconstructed using NIS Elements software v.4.20 (Nikon) and used for counting aperture number.

For imaging cells of the developing pollen lineage, anthers were dissected out of stage 9 flower buds and placed into a small drop of Vectashield antifade solution supplemented with 0.02% Calcofluor White and 5 μg/mL membrane stain CellMask Deep Red. Cells in the pollen lineage were released by applying gentle pressure to the coverslip placed over the anthers. To obtain fluorescence signals, the following excitation/emission settings were used: RFP, 561 nm/580–630 nm; YFP, 514 nm/522–555 nm; Calcofluor White, 405 nm/424–475 nm; CellMask Deep Red, 640 nm/663–738 nm. Z-stacks of tetrads were obtained with a step size of 500 nm and 3D reconstructed using NIS Elements v.4.20 (Nikon).

To compare the YFP fluorescence intensity in three different lines of *mcr MCRpr:gMCR-YFP* or *mcr elmod_a EApr:gELMOD_A-YFP*, tetrads were prepared simultaneously and imaged on the same day under identical acquisition settings on Nikon A1+ confocal microscope. The mean YFP signal

intensities in nucleoplasm and cytoplasm of tetrads (n ≥ 15) were separately measured with the help of the region of interest (ROI) statistics function in NIS Elements v.4.20 (Nikon). For each tetrad, a single optical section showing both nucleoplasm and cytoplasm was selected and analyzed.

## Sequence retrieval and phylogenetic analysis of the plant ELMOD family

ELMOD family members in *Arabidopsis* have the following accession numbers: *ELMOD_A, At3g60260; MCR, At2g44770; ELMOD_C, At1g67400; ELMOD_D, At3g43400; ELMOD_E, At1g03620; ELMOD_F, At3g03610*. The phylogenetic tree of *Arabidopsis* ELMOD proteins in *Figure 3E* was built using the neighbor-joining (NJ) algorithm of MEGA X (*Kumar et al., 2018*), with bootstrap support calculated for 1000 replicates.

Sequences of ELMOD proteins from species across the plant kingdom were retrieved from the Phytozome v.12 database (https://phytozome.jgi.doe.gov/pz/portal.html) and the 1000 Plants (1KP) database (https://db.cngb.org/onekp/, last accessed in May 2020; *Wickett et al., 2014*). MCR protein sequence was used as a query for an online BLASTP search of these databases with default parameters. The protein sequences with the E-value ≤1e-10, sequence identity ≥30%, and Bit-Score ≥ 60 were identified as ELMODs and further confirmed by a local BLASTP search using each of the other *Arabidopsis* ELMODs as a query. In the cases when two or more proteins were potentially translated from the same gene, the one providing the best match with the query was selected. In total, 561 ELMOD protein sequences from 178 representative species belonging to eudicots (36 species/195 sequences), monocots (14/94), magnoliids (20/64), basal angiosperms (5/13), gymnosperms (17/62), ferns (17/44), lycophytes (20/29), bryophytes (37/47; including 18 sequences from 15 liverworts, 6 sequences from 5 hornworts, and 23 sequences from 17 mosses), and green algae (12/13) were retrieved and used for phylogenetic analysis.

Multiple sequence alignment was performed using MAFFT v7.017 (*Katoh and Standley, 2013*; *Katoh et al., 2002*) with the L-INS-i algorithm and default parameters. Sites with greater than 20% gaps were trimmed by TrimAl (*Capella-Gutiérrez et al., 2009*) and manually inspected for overhangs. ModelFinder (*Kalyaanamoorthy et al., 2017*; accessed through IQ-TREE [*Nguyen et al., 2015*]) was run to find the best-fit amino acid substitution model. The alignment in *Figure 3—figure supplement 1* was visualized with Espript3.0 (*Gouet et al., 1999*). Phylogenetic trees were constructed using IQ-TREE with the maximum likelihood (ML) method, SH-aLRT test, and ultrafast bootstrap with 1000 replicates. For the tree on *Figure 8A*, containing sequences from across the plant kingdom, 267 sequences were used, including all sequences retrieved from green algae, bryophytes, lycophytes, ferns, gymnosperms, and basal angiosperms, as well as 24 sequences from magnoliids, 19 sequences from three monocots, and 16 sequences from three eudicots. For the tree on *Figure 8—figure supplement 1*, containing only angiosperm sequences, we used all 366 sequences retrieved for this group. Phylogenetic trees were visualized in iTOL v.5 (*Letunic and Bork, 2021*) and can be accessed at http://itol.embl.de/shared/Zhou3117.

## Expression pattern analysis of the *Arabidopsis ELMOD* genes

RNA-seq data for different tissues/developmental stages of six *Arabidopsis ELMOD* genes were obtained from the TRAVA database (http://travadb.org/; *Klepikova et al., 2016*). The 'Raw Norm' option was chosen for read counts, and default settings were used for all other options. The retrieved RNA-seq data were presented as a bubble heatmap using TBtools (*Chen et al., 2020*).

## Quantification and statistical analysis

Quantification of aperture numbers and YFP signal was done with NIS Elements v.4.20 software (Nikon). For each line, the aperture number of 160 pollen grains from at least three different plants was counted and the mean YFP fluorescence of at least 15 tetrads from the same plants was measured. Graphs were generated using Microsoft Excel or Origin version 2018 software. Binary comparisons were performed using a two-tailed Student's t-test in Microsoft Excel; results with the p values below 0.05 were judged significantly different. The p values are represented as (***p<0.001), (**p<0.01), *p<0.05. All error bars represent standard deviation (SD). For all boxplots, the box defines the first and third quartile, the central line depicts the median, and the small square in the box represents the mean value. Whiskers extend to minimum and maximum values. Outliers are indicated as *. Different

shapes show individual samples. Details of statistical analysis, number of quantified entities (n), and measures of dispersion can be found in the corresponding figure legends.

## Acknowledgements

Funding for this project was provided by the US National Science Foundation (MCB-1817835, awarded to AAD). We also acknowledge the support of the OSU Mayers Undergraduate Summer Research Scholarship to PA, the NSF-REU supplement funding to PA and AH, and the iCAPS internship from the OSU Center for Applied Sciences to AH.

## Additional information

### Funding

| Funder | Grant reference number | Author |
| --- | --- | --- |
| National Science Foundation | MCB-1817835 | Anna A Dobritsa |
| Ohio State University | Mayers Undergraduate Summer Research Scholarship | Prativa Amom |
| National Science Foundation-Research Experience for Undergraduates Supplement funding | MCB-1517511 | Prativa Amom Adam Helton Anna A Dobritsa |
| Ohio State University Center for Applied Plant Sciences | iCAPS internship | Adam Helton |

The funders had no role in study design, data collection and interpretation, or the decision to submit the work for publication.

### Author contributions

Yuan Zhou, Conceptualization, Formal analysis, Investigation, Methodology, Visualization, Writing – original draft, Writing – review and editing; Prativa Amom, Byung Ha Lee, Formal analysis, Investigation; Sarah H Reeder, Investigation, Writing – review and editing; Adam Helton, Investigation; Anna A Dobritsa, Conceptualization, Formal analysis, Funding acquisition, Investigation, Methodology, Project administration, Supervision, Visualization, Writing – original draft, Writing – review and editing

### Author ORCIDs

Yuan Zhou http://orcid.org/0000-0002-3598-958X
Adam Helton http://orcid.org/0000-0003-2948-2368
Anna A Dobritsa http://orcid.org/0000-0003-2987-1718

### Decision letter and Author response

Decision letter https://doi.org/10.7554/eLife.71061.sa1
Author response https://doi.org/10.7554/eLife.71061.sa2

## Additional files

### Supplementary files

• Supplementary file 1. Primers, molecular markers, and mutants/transgenic lines used in this study.

• Transparent reporting form

### Data availability

All data generated or analysed during this study are included in the manuscript and supporting files. Source data files have been provided for Figure 1I; Figure 3E; Figures 7C, D and F; Figure 8 and Figure

8-figure supplement 1; Figure 9E and Figure 9-figure supplement 2E; Figure 10 and Figure10-figure supplement 1C.

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
