## [Decision Letter]

**Acceptance summary:**

The authors identify a role in Arabidopsis pollen aperture formation for members of the ELMOD family of proteins. They use genetics, transgenic constructs, and site-directed mutagenesis to pinpoint important residues for protein function in this process. The authors thoroughly addressed the reviewer comments.

**Decision letter after peer review:**

Thank you for submitting your article "Members of the ELMOD protein family specify formation of distinct aperture domains on the Arabidopsis pollen surface" for consideration by *eLife*. Your article has been reviewed by 3 peer reviewers, including Sheila McCormick as the Reviewing Editor and Reviewer #1, and the evaluation has been overseen by Jürgen Kleine-Vehn as the Senior Editor. The following individual involved in review of your submission has agreed to reveal their identity: Sharon A Kessler (Reviewer #2).

Essential revisions:

1) Address the issue of hypomorphic T-DNA alleles in the Discussion.

2) If possible, provide a more thorough quantification of round vs furrow in the complementation experiments.

3) Consider including information about the fertility of the mutants, to enrich the biological context.

*Reviewer #1 (Recommendations for the authors):*

Lines 164-167. Odd that the T-DNA mutations were hypomorphic – this needs to be addressed in the discussion.

Line 194 – the more ancient of what? ELMOD and larger ELMO? What is the D supposed to designate, the word domain? Please clarify.

Line 396, a bit confusing, consider re-ordering this section. At first it is not clear why is it notable that no other plants have an Asp at the Gly129 site. Having an Asp is a mutant, right? Later you say that plants have either Cys, Gly or Ala.

Line 492-3, I would tone down this statement.

The discussion is too speculative about things that are not addressed in this paper.

And why the sentences about Pedicularis? I would delete, i.e. lines 562-563 are complete speculation.

Lines 590-592 don't add much; same comment for lines 607-609, same for lines 610-615.

*Reviewer #2 (Recommendations for the authors):*

The manuscript is clearly written and a pleasure to read. I just have few suggestions for improving the paper.

1. The ploidy and microspore arrangement data in Figures 1 and 1-S2 are interesting, but they are not linked to any of the other data in the paper and not mentioned at all in the discussion. This data could be removed to streamline the paper.

2. The ELMOD_A CRISPR description in the Results section is difficult to follow. The description implied that the lines used for the mcr complementation studies had undefined CRISPR-induced mutations. These lines were more clearly described in the methods-some of this description should be moved to the results and the exact effect of the CRISPR allele in Figure 4 needs to be more clearly defined. The single A insertion is near an exon/intron junction-does it affect splicing? Is it actually a knock-out allele, or does it retain some function? This is important for the claim that single elmod_a mutants don't have a phenotype-if this is just a knockdown allele then a true knockout could have a more severe phenotype, even though both would be complemented with a wild-type transgene.

3. It is not clear why "furrowed and round" apertures were lumped together in one category in Figure 10 D-F. Dividing them into 2 categories would allow the authors to compare the complementation of aperture shape with the ELMOD_E variants (are they mostly round or mostly furrowed, or is it even?)

4. Lines 395-398: I don't think it is relevant to look specifically for a GLY129ASP change in the other ELMODs. This specific mutation came from an EMS mutagenesis-why would it be expected to occur naturally?

5. Lines 417-431: The conclusion that ELMOD_D is a pseudogene may be premature since it is expressed. The authors should add the alternative explanation that this gene may have acquired a new function.

6. As I looked at the beautiful pollen aperture phenotypes throughout the manuscript, I wondered if the aperture area was conserved in the different mutant/ ectopic expression backgrounds. Having 1 aperture that extends around the circumference of the pollen grain is obviously not the same as having 1 small, round aperture, but is it similar in area to 3 furrows? The ELMOD_E lines with round apertures also tended to have more apertures-is the total aperture area regulated by some other mechanism? This is just a comment, not a suggestion that needs to be addressed with more experiments for this publication.

*Reviewer #3 (Recommendations for the authors):*

1. Fig. 1F,G, suggest using a different colored arrowheads, e.g. red on a greenish yellow background and signal, not yellow as they currently are.

2. Up till point 2 in the previous section, and L152, what I missed is the pollen tube emergence and fertility phenotype of these and previously characterized mutants. Later on, L239 referred to a double mcr elmod_a double mutant; again a note on fertility would be good. It would enrich the biological context to have related these in the Introduction for the inp, d6pk3, mcr-1 mutants, and mention it for the newly created mutants (even when without impact).

3. L163-end, can you provide insight on why the T-DNA insertions are hypomorphic, whereas mcr1 is totally penetrant (0% normal), what about mcr2-4?

4. Fig. 4; it would be nice to have indicated the percentage of defective pollen, especially in the homozygous heterozygous combinations.

5. I find that the detail descriptions of elmod e,d,f in Fig. 9, while the genetic and morphological results were hard-earned, they are distracting. The authors can very easily summarize them in one or two sentences and let the results be in Supplement, and move on to focus on the ELMOD E studies. Also L475-477 is cumbersome; is it the same as in the original line 7-2/mcr?

6. Fig. 6, N121 C129 are not shown in blue as indicated in the legend.

---

## [Author Response]

Essential revisions:1) Address the issue of hypomorphic T-DNA alleles in the Discussion.

All three T-DNA alleles have their insertion sites inside introns, suggesting these alleles might be knockdown mutants that still retain some MCR activity. We have now added this statement to the Results. We do not think that this point requires to be further addressed in the Discussion, since T-DNA insertions in introns that do not cause a complete gene inactivation are not too unusual (please see our response to Reviewer 1 for references and additional discussion on this point).

2) If possible, provide a more thorough quantification of round vs furrow in the complementation experiments.

We have now quantified these results based on the images of representative pollen grains and show the data in Figure 10—figure supplement 1C.

3) Consider including information about the fertility of the mutants, to enrich the biological context.

All the mutants have normal fertility, like all the other Arabidopsis aperture mutants we have been working with, including mutants that produce no apertures. We have now included this information in the Results (lines 88-91).

Reviewer #1 (Recommendations for the authors):Lines 164-167. Odd that the T-DNA mutations were hypomorphic – this needs to be addressed in the discussion.

We do not think that the observation that T-DNA insertion alleles are hypomorphic is too surprising. All three T-DNA insertion mutants have their insertions sites within introns. Furthermore, in *mcr-7*, which produces the largest proportion of pollen with wild-type apertures (~20%), the T-DNA insertion lies before the beginning of the coding sequence. It is thus probable that these mutations create knockdown, rather than knockout, alleles, which does happen with intron-based T-DNA insertions. See, for example, the following references:

Wang, Y. H. (2008) How effective is T-DNA insertional mutagenesis in Arabidopsis. J Bioch Tech 1, 11-20

Sato, H., Shibata, F., Murata, M. (2005) Characterization of a Mis12 homologue in *Arabidopsis thaliana*. Chromosome Res 13, 827–834. https://doi.org/10.1007/s10577-005-1016-3

Chuang, H., Zhang, W., Gray, W. M. (2004) Arabidopsis *ETA2*, an apparent ortholog of the human cullin-interacting protein CAND1, is required for auxin responses mediated by the SCF^TIR1^ ubiquitin ligase. Plant Cell 16, 1883–1897. https://doi.org/10.1105/tpc.021923

Verelst, W., Saedler, H., Münster, T. (2007) MIKC* MADS-protein complexes bind motifs enriched in the proximal region of late pollen-specific Arabidopsis promoters, Plant Physiol 143, 447–460, https://doi.org/10.1104/pp.106.089805

Jones, M.A., Raymond, M.J. and Smirnoff, N. (2006), Analysis of the root-hair morphogenesis transcriptome reveals the molecular identity of six genes with roles in root-hair development in Arabidopsis. Plant J 45, 83-100. https://doi.org/10.1111/j.1365-313X.2005.02609.x

Imai, A., Matsuyama, T., Hanzawa, Y., Akiyama, T., Tamaoki, M., Saji, H., Shirano, Y., Kato, T., Hayashi, H., Shibata, D., Tabata, S., Komeda, Y., Takahashi, T. (2004) Spermidine synthase genes are essential for survival of Arabidopsis, Plant Physiol 135, 1565–1573, https://doi.org/10.1104/pp.104.041699

Collados Rodríguez, M., Wawrzyńska, A., Sirko, A. (2014) Intronic T-DNA insertion in Arabidopsis *NBR1* conditionally affects wild-type transcript level, Plant Signal Behav, 9:12, e975659,DOI: 10.4161/15592324.2014.975659

Sandhu, K. S., Koirala, P. S., Neff, M. M. (2013) The *ben1-1* brassinosteroid-catabolism mutation is unstable due to epigenetic modifications of the intronic T-DNA insertion, G3, 3, 1587–1595, https://doi.org/10.1534/g3.113.006353

The main purpose for our use of these alleles was to find out if they would also have the *mcr* aperture phenotype, which they all did. This served as one more piece of evidence that we had identified the correct gene. We have now added a statement about the intronic location of T-DNA insertions to the Results (line 165), but we do not think this point requires a further discussion.

Line 194 – the more ancient of what? ELMOD and larger ELMO? What is the D supposed to designate, the word domain? Please clarify.

This nomenclature is already established based on the animal protein studies, and for this reason we would like to adhere to it, but it is certainly confusing. The D in ELMOD represents ‘domain’, and the family and its members are typically referred to simply as ‘ELMOD’ to differentiate them from members of the ELMO family (which also have the ELMO domain). Compared to the ELMO family, the ELMOD family is believed to be the more ancient. We have now clarified this point (lines 195-197).

Line 396, a bit confusing, consider re-ordering this section. At first it is not clear why is it notable that no other plants have an Asp at the Gly129 site. Having an Asp is a mutant, right? Later you say that plants have either Cys, Gly or Ala.

In total, we retrieved 561 ELMOD sequences from across the plant kingdom. Among them, only one sequence, which happened to be AtELMOD_D from Arabidopsis, a possible pseudogene, was found to have an Asp at the position 129. The other 560 ELMOD sequences have either Cys, Gly or Ala at the position 129. This analysis suggested that Asp at this position is not tolerated by natural selection and might be detrimental for all plant ELMOD proteins. In the MCR protein of the *mcr-2* mutant, Gly129 is replaced with Asp, which appears to make this protein non-functional. Later in the manuscript we explain that in angiosperms we found a very strong correlation between an amino acid occupying position 129 and a clade with which each protein sequence clustered, suggesting that this position could be important for protein function as well as for functional diversification among the ELMOD clades.

We have now modified this paragraph (lines 395-400), transferring the emphasis from Gly129 to the appearance of Asp, which represents a highly unusual amino acid for this position. The discussion of the amino acids that are normally found at this position is left until the next paragraph.

Line 492-3, I would tone down this statement.

We have replaced this statement with the following: “Together, these data support the idea that high level of MCR can counteract the neomorphic activity of ELMOD_E and suggest that MCR and ELMOD_E may compete for the same interactors”.

The discussion is too speculative about things that are not addressed in this paper.

We have modified portions of the Discussion, but we also use it as a place for putting our findings into a larger context and presenting some ideas for future studies.

And why the sentences about Pedicularis? I would delete, i.e. lines 562-563 are complete speculation.

We have modified these sentences to make it clearer that we are speculating here, but we still would like to keep them. How the diversity of aperture patterns is generated is one of the most fascinating questions in this field. Mentioning the aperture patterns in Pedicularis, some of which resemble the *mcr* phenotype in Arabidopsis, allows us to connect the mutant phenotype we discovered in a laboratory model organism to a phenomenon that occurs in nature and highlights variations of aperture patterns within a single genus. By suggesting that variations in aperture patterns in Pedicularis could be caused by variations in ELMOD proteins or their regulators or interactors, we formulate a potentially testable hypothesis.

Lines 590-592 don't add much; same comment for lines 607-609, same for lines 610-615.

We believe that the sentences in lines 590-592 (lines 588-590 in the revision) make an important point. The fact that in most analyzed angiosperm species the A/B clade had more than one member suggest that there could be a selective pressure to maintain redundancy of that protein activity, hinting at the importance of the process(es) in which these proteins are involved.

We removed lines 607-609, but we kept the lines 610-615 (lines 605-611 in the revision) since we think it is important to mention the possibility that plant ELMODs might be doing something different from what they are doing in animals.

Reviewer #2 (Recommendations for the authors):The manuscript is clearly written and a pleasure to read. I just have few suggestions for improving the paper.1. The ploidy and microspore arrangement data in Figures 1 and 1-S2 are interesting, but they are not linked to any of the other data in the paper and not mentioned at all in the discussion. This data could be removed to streamline the paper.

We would prefer to keep this part because it provides important observations related to the mechanism of aperture pattern formation. It shows that the reducing effect of the *mcr* mutation on aperture number is quite general, since it is manifested across different levels of ploidy and post-meiotic microspore arrangement, which we previously showed to be important factors in aperture patterning. How higher ploidy causes an increase in aperture number is still unclear, but an increased expression of aperture factors due to higher gene dosage is one possibility. This would be consistent with the data we present later in the manuscript showing that apertures are sensitive to the levels of MCR and ELMOD_A expression. To make this part less ‘stand-alone’, we now refer to this result in the Discussion (lines 556-559).

2. The ELMOD_A CRISPR description in the Results section is difficult to follow. The description implied that the lines used for the mcr complementation studies had undefined CRISPR-induced mutations. These lines were more clearly described in the methods-some of this description should be moved to the results and the exact effect of the CRISPR allele in Figure 4 needs to be more clearly defined. The single A insertion is near an exon/intron junction-does it affect splicing? Is it actually a knock-out allele, or does it retain some function? This is important for the claim that single elmod_a mutants don't have a phenotype-if this is just a knockdown allele then a true knockout could have a more severe phenotype, even though both would be complemented with a wild-type transgene.

The CRISPR mutant of ELMOD_A is expected to be a knockout. The mutation is an insertion of a single A nucleotide early in the coding sequence (between codons 64 and 65), just before the beginning of the sequence encoding the ELMO domain and three nucleotides before the beginning of the second intron (the inserted nucleotide is at the position -4 relative to the intron). This insertion creates a frame shift that would lead to a greatly truncated protein independently of whether it affects the splicing of that intron. With this insertion, the translation machinery will encounter premature stop codons in all possible scenarios: (a) if splicing occurs normally; (b) if the second intron is not spliced; (c) if the next GT site is used as a donor site for splicing. (Still, the splicing of this intron is unlikely to be affected in this case as only the last two positions (-1 and -2) at the end an exon are typically important for pairing with snRNAs, while other exonic splicing signals lie much further interior within exons (Wilkinson et al. (2020) Ann. Rev. Biochem. 89:359-388; Fu et al. (2004) Cell 119: 736-738)).

We should also mention that in addition to this allele, which was used for crossing with the *mcr* mutation and all the subsequent work, we generated a second CRISPR allele of ELMOD_A which has a single nucleotide deletion (DT) in the last nucleotide of the codon 64. This mutation causes a different frame shift and is also expected to result in very early termination of translation in any splicing scenario, yet the mutant also produced pollen with normal apertures.

We have modified the part in the Results describing the *elmod_a* CRISPR alleles to clarify the nature of these mutations (lines 212-217). We have also changed the illustration of the *elmod_a* CRISPR allele on Figure 4A to indicate more clearly that a frame shift occurred. (We have similarly changed the illustrations for *elmod_e* CRISPR alleles on Figure 9—figure supplement 1A.)

3. It is not clear why "furrowed and round" apertures were lumped together in one category in Figure 10 D-F. Dividing them into 2 categories would allow the authors to compare the complementation of aperture shape with the ELMOD_E variants (are they mostly round or mostly furrowed, or is it even?)

We have now quantified these results based on images of representative pollen grains from across the multiple T_1_ lines expressing the same construct and show the data in Figure 10—figure supplement 1C. In cases when three apertures form, they tend to be furrow-like, whereas when ≥4 apertures form, they tend to be round.

4. Lines 395-398: I don't think it is relevant to look specifically for a GLY129ASP change in the other ELMODs. This specific mutation came from an EMS mutagenesis-why would it be expected to occur naturally?

To better understand how significant a mutation is for protein function, it would certainly make sense to check whether an amino acid affected by that mutation normally shows strong conservation, whether the change is substantial from a biochemical perspective, and whether a similar substitution might exist in related proteins in nature. To do this, it helps to look at the amino acids occupying the same position in orthologs and paralogs of the studied protein.

In the case of the Gly129Asp mutation in *mcr-2*, we found that the appearance of Asp represents a very unusual change from the evolutionary perspective: only one sequence out of the 561 analyzed contained an Asp at that position, and as explained below, we suspect that sequence actually corresponds to a pseudogene. The position 129 is quite conserved, with only three amino acids normally found at that position across the plant ELMOD family and nearly all A/B proteins having a Gly there. Finally, from the biochemical perspective, an appearance of the negatively charged Asp instead of Gly (or Ala and Cys that are also found at this position) represents a substantial change.

We have modified this paragraph (lines 395-400) to emphasize how unusual the appearance of Asp at this position is (with the implication that such change could be sufficient to inactivate the protein). The discussion of the amino acids normally occupying this position is now left until the next paragraph.

5. Lines 417-431: The conclusion that ELMOD_D is a pseudogene may be premature since it is expressed. The authors should add the alternative explanation that this gene may have acquired a new function.

Although it is typically difficult to prove that a sequence behaves as a pseudogene in the absence of early stop codons or other mutations that would clearly disrupt its function (frame shifts, large insertions or deletions), we think that the combined evidence for ELMOD_D being a likely pseudogene is quite strong:

1) The mRNA expression levels of *ELMOD_D* are extremely low (no more than 2 normalized read counts in any tissue listed in the RNA-seq database TRAVA http://travadb.org/browse/DeSeq/At3g43400/RawNorm/AvNorm/Color=RCount/) (Figure 3E). (Since the wording describing the nearly complete lack of ELMOD_D expression might have been a bit confusing in the previous version of the manuscript, we have slightly modified this part (line 431)).

2) The YFP-tagged ELMOD_D protein, driven by the *MCR* regulatory elements, could not be detected (Figure 9—figure supplement 2H-H’).

3) The predicted protein is much shorter than the other members of the ELMOD family, with multiple regions missing, including some conserved portions of the ELMO domain (Figure 3—figure supplement 1).

4) As explained above, ELMOD_D is the only ELMOD protein out of the 561 examined which contains an Asp at position 129. A mutation that creates Asp at the same position in the MCR protein leads to protein inactivation. In addition, arginines found in ELMOD_D at the two positions near the GAP region (corresponding to Gly119 and Leu138 in MCR) also represent highly unusual and biochemically significant changes that are not found in any other ELMOD protein.

5) Other plants, including close relatives of Arabidopsis, lack orthologs of ELMOD_D.

Although each of these reasons, on their own, would not be sufficient to suggest that ELMOD_D is a pseudogene, we think that together they provide a good argument in favor of this possibility, which we still describe as a hypothesis, not a fact.

6. As I looked at the beautiful pollen aperture phenotypes throughout the manuscript, I wondered if the aperture area was conserved in the different mutant/ ectopic expression backgrounds. Having 1 aperture that extends around the circumference of the pollen grain is obviously not the same as having 1 small, round aperture, but is it similar in area to 3 furrows? The ELMOD_E lines with round apertures also tended to have more apertures-is the total aperture area regulated by some other mechanism? This is just a comment, not a suggestion that needs to be addressed with more experiments for this publication.

It’s an interesting idea! In general, the more furrow-shaped apertures develop on the pollen surface, the shorter they tend to be, so this hypothesis would make sense. Such study, however, is likely to be tricky, since in order to see apertures on the entire pollen surface, the normally spherical pollen grains have to be flattened. As a result, the apparent shape and size of apertures might be somewhat distorted, which may affect measurements and conclusions.

Reviewer #3 (Recommendations for the authors):1. Fig. 1F,G, suggest using a different colored arrowheads, e.g. red on a greenish yellow background and signal, not yellow as they currently are.

We have changed the color of arrows to red in Fig. 1F, 1G (and in the similar Fig. 2A).

2. Up till point 2 in the previous section, and L152, what I missed is the pollen tube emergence and fertility phenotype of these and previously characterized mutants. Later on, L239 referred to a double mcr elmod_a double mutant; again a note on fertility would be good. It would enrich the biological context to have related these in the Introduction for the inp, d6pk3, mcr-1 mutants, and mention it for the newly created mutants (even when without impact).

All the described mutants have no noticeable fertility defects and produce normal seed sets. The normal fertility in *mcr* and other mutants studied here is consistent with the normal fertility of the previously characterized pollen aperture mutants in Arabidopsis, including the ones that completely lack apertures (*inp1*, *inp2*) (Dobritsa et al. (2011) Plant Phys. 157:947-970; Dobritsa and Coerper (2012) Plant Cell 24:4452-4464; Lee et al. (2021) Nat. Plants 7:966-978). Additionally, it has been previously shown that pollen tubes in Arabidopsis and many other species of Brassicaceae have an ability to emerge through exine and not through the aperture sites (Edlund et al. (2016) Am. J. Bot. 103:1006-1019). For these reasons, observing normal fertility in the *mcr* mutants was not surprising. Thus, we have not specifically looked at the pollen tube emergence in the *mcr* mutants, but their ability to set a large number of seeds indicates that their pollen tubes do not have problems with emerging.

We have now added a statement mentioning that, like other aperture mutants in Arabidopsis, *mcr* mutants exhibit no noticeable fertility problems (lines 88-91). For the reasons mentioned above, we do not think that we need to indicate this for all the other *mcr* combinations created in this study.

3. L163-end, can you provide insight on why the T-DNA insertions are hypomorphic, whereas mcr1 is totally penetrant (0% normal), what about mcr2-4?

As described in our response to the question from the Reviewer 1, we think that because the insertion sites of all the T-DNA mutants are in introns, these mutants represent knockdown alleles that retain partial activity. Pollen in *mcr-*2 through *mcr* -4 only has *mcr* aperture phenotype.

4. Fig. 4; it would be nice to have indicated the percentage of defective pollen, especially in the homozygous heterozygous combinations.

All the mutants in Fig. 4 (those not carrying *MCR* or *ELMOD_A* transgenes) produced viable pollen grains with the indicated aperture phenotypes. In each case, the phenotypes were fully penetrant. The mutants expressing transgenes showed some variations in aperture number, with some transgenic lines producing more than the expected number of apertures, but we describe this later in the manuscript.

5. I find that the detail descriptions of elmod e,d,f in Fig. 9, while the genetic and morphological results were hard-earned, they are distracting. The authors can very easily summarize them in one or two sentences and let the results be in Supplement, and move on to focus on the ELMOD E studies. Also L475-477 is cumbersome; is it the same as in the original line 7-2/mcr?

We have somewhat shortened the two paragraphs dealing with the *C*, *D*, and *F* genes, and rearranged the panels in Figure 9 – Supplemental Fig. 2 to follow the narrative. We have simplified the sentence in L475-477 and combined it with a previous sentence. The new sentence (lines 467-470) reads: “Yet, in the F_1_ progeny of the cross with the high expressor 7-2, furrow aperture morphology was restored (Figure 9K–9K’), although these plants produced less pollen with a high number of furrows (>4) compared to the original 7-2 plants (Figure 9L).”

6. Fig. 6, N121 C129 are not shown in blue as indicated in the legend.

These two amino acids of AtELMOD_E were in fact indicated in blue in the previous version, but to make them more noticeable, we have now indicated them by red squares.